# Layer-wise Adaptive Model Aggregation for Scalable Federated Learning

## Abstract

In Federated Learning, a common approach for aggregating local models across clients is periodic averaging of the full model parameters. It is, however, known that different layers of neural networks can have a different degree of model discrepancy across the clients. The conventional full aggregation scheme does not consider such a difference and synchronizes the whole model parameters at once, resulting in inefficient network bandwidth consumption. Aggregating the parameters that are similar across the clients does not make meaningful training progress while increasing the communication cost. We propose FedLAMA, a layer-wise model aggregation scheme for scalable Federated Learning. FedLAMA adaptively adjusts the aggregation interval in a layer-wise manner, jointly considering the model discrepancy and the communication cost. The layer-wise aggregation method enables to finely control the aggregation interval to relax the aggregation frequency without a significant impact on the model accuracy. Our empirical study shows that FedLAMA reduces the communication cost by up to $60\%$ for IID data and $70\%$ for non-IID data while achieving a comparable accuracy to FedAvg.

## 1 Introduction

In Federated Learning, periodic full model aggregation is the most common approach for aggregating local models across clients. Many Federated Learning algorithms, such as FedAvg (McMahan et al. (2017)), FedProx (Li et al. (2018)), FedNova (Wang et al. (2020)), and SCAFFOLD (Karimireddy et al. (2020)), assume the underlying periodic full aggregation scheme. However, it has been observed that the magnitude of gradients can be significantly different across the layers of neural networks (You et al. (2019)). That is, all the layers can have a different degree of model discrepancy. The periodic full aggregation scheme does not consider such a difference and synchronizes the entire model parameters at once. Aggregating the parameters that are similar across all the clients does not make meaningful training progress while increasing the communication cost. Considering the limited network bandwidth in usual Federated Learning environments, such an inefficient network bandwidth consumption can significantly harm the scalability of Federated Learning applications.

Many researchers have put much effort into addressing the expensive communication issue. Adaptive model aggregation methods adjust the aggregation interval to reduce the total communication cost (Wang & Joshi (2018); Haddadpour et al. (2019)). Gradient (model) compression (Alistarh et al. (2018); Albasyoni et al. (2020)), sparsification (Wangni et al. (2017); Wang et al. (2018)), low-rank approximation (Vogels et al. (2020); Wang et al. (2021)), and quantization (Alistarh et al. (2017); Wen et al. (2017); Reisizadeh et al. (2020)) techniques directly reduce the local data size. Employing heterogeneous model architectures across clients is also a communication-efficient approach (Diao et al. (2020)). While all these works effectively tackle the expensive communication issue from different angles, they commonly assume the underlying periodic full model aggregation.

To break such a convention of periodic full model aggregation, we propose FedLAMA, a novel layer-wise adaptive model aggregation scheme for scalable and accurate Federated Learning. FedLAMA first prioritizes all the layers based on their contributions to the total model discrepancy. We present a metric for estimating the layer-wise degree of model discrepancy at run-time. The aggregation intervals are adjusted based on the layer-wise model discrepancy such that the layers with a smaller degree of model discrepancy is assigned with a longer aggregation interval than the other layers. The above steps are repeatedly performed once the entire model is synchronized once.

Our focus is on how to relax the model aggregation frequency at each layer, jointly considering the communication efficiency and the impact on the convergence properties of federated optimization. By adjusting the aggregation interval based on the layer-wise model discrepancy, the local models can be effectively synchronized while reducing the number of communications at each layer. The model accuracy is marginally affected since the intervals are increased only at the layers that have the smallest contribution to the total model discrepancy. Our empirical study demonstrates that FedLAMA automatically finds the interval settings that make a practical trade-off between the communication cost and the model accuracy. We also provide a theoretical convergence analysis of FedLAMA for smooth and non-convex problems under non-IID data settings.

We evaluate the performance of FedLAMA across three representative image classification benchmark datasets: CIFAR-10 (Krizhevsky et al. (2009)), CIFAR-100, and Federated Extended MNIST (Cohen et al. (2017)). Our experimental results deliver novel insights on how to aggregate the local models efficiently consuming the network bandwidth. Given a fixed number of training iterations, as the aggregation interval increases, FedLAMA reduces the communication cost by up to $60\%$ under IID data settings and $70\%$ under non-IID data settings, while having only a marginal accuracy drop.

## 2   RELATED WORKS

**Compression Methods** – The communication-efficient global model update methods can be categorized into two groups: *structured* update and *sketched* update (Konečnỳ et al. (2016)). The *structured* update indicates the methods that enforce a pre-defined fixed structure of the local updates, such as low-rank approximation and random mask methods. The *sketched* update indicates the methods that post-process the local updates via compression, sparsification, or quantization. Both directions are well studied and have shown successful results (Alistarh et al. (2018); Albasyoni et al. (2020); Wangni et al. (2017); Wang et al. (2018); Vogels et al. (2020); Wang et al. (2021); Alistarh et al. (2017); Wen et al. (2017); Reisizadeh et al. (2020)). The common principle behind these methods is that the local updates can be replaced with a different data representation with a smaller size.

These compression methods can be independently applied to our layer-wise aggregation scheme such that the each layer's local update is compressed before being aggregated. Since our focus is on adjusting the aggregation frequency rather than changing the data representation, we do not directly compare the performance between these two approaches. We leave harmonizing the layer-wise aggregation scheme and a variety of compression methods as a promising future work.

**Similarity Scores** – Canonical Correlation Analysis (CCA) methods are proposed to estimate the representational similarity across different models (Raghu et al. (2017); Morcos et al. (2018)). Centered Kernel Alignment (CKA) is an improved extension of CCA (Kornblith et al. (2019)). While these methods effectively quantify the degree of similarity, they commonly require expensive computations. For example, SVCCA performs singular vector decomposition of the model and CKA computes Hilbert-Schmidt Independence Criterion multiple times (Gretton et al. (2005)). In addition, the representational similarity does not deliver any information regarding the gradient difference that is strongly related to the convergence property. We will propose a practical metric for estimating the layer-wise model discrepancy, which is cheap enough to be used at run-time.

**Layer-wise Model Freezing** – Layer freezing (dropping) is the representative layer-wise technique for neural network training (Brock et al. (2017); Kumar et al. (2019); Zhang & He (2020); Goutam et al. (2020)). All these methods commonly stop updating the parameters of the layers in a bottom-up direction. These empirical techniques are supported by the analysis presented in (Raghu et al. (2017)). Since the layers converge from the input-side sequentially, the layer-wise freezing can reduce the training time without strongly affecting the accuracy. These previous works clearly demonstrate the advantages of processing individual layers separately.

## 3   BACKGROUND

**Federated Optimization** – We consider federated optimization problems as follows.

$$\min_{\mathbf{x} \in \mathbb{R}^d} \left[ F(\mathbf{x}) := \sum_{i=1}^{m} p_i F_i(\mathbf{x}) \right], \tag{1}$$

---

**Algorithm 1:** FedLAMA: Federated Layer-wise Adaptive Model Aggregation.

---

**Input:** $\tau'$: base aggregation interval, $\phi$: interval increasing factor, $p_i, i \in \{1, \cdots, m\}$.

1   $\tau_l \leftarrow \tau', \forall l \in \{1, \cdots, L\}$;

2   **for** $k = 1$ *to* $K$ **do**

3      SGD step: $\mathbf{x}_k^i = \mathbf{x}_{k-1}^i - \eta \nabla f(w_{k-1}^i, \xi_k)$;

4      **for** $l = 1$ *to* $L$ **do**

5         **if** $k \bmod \tau_l$ *is* 0 **then**

6            Synchronize layer $l$: $\mathbf{u}_{(l,k)} \leftarrow \sum_{i=1}^m p_i \mathbf{x}_{(l,k)}^i$;

7            $d_l \leftarrow \sum_{i=1}^m \left( p_i \| \mathbf{u}_{(l,k)} - \mathbf{x}_{(l,k)}^i \|^2 \right) / (\tau_l(\dim(\mathbf{u}_{(l,k)})))$ ;

8      **if** $k \bmod \phi\tau'$ *is* 0 **then**

9         Adjust aggregation interval at all $L$ layers (Algorithm 2).;

10 **Output:** $\mathbf{u}_K$;

---

where $p_i = n_i/n$ is the ratio of local data to the total dataset, and $F_i(\mathbf{x}) = \frac{1}{n_i} \sum_{\xi \in \mathcal{D}} f_i(\mathbf{x}, \xi)$ is the local objective function of client $i$. $n$ is the global dataset size and $n_i$ is the local dataset size.

FedAvg is a basic algorithm that solves the above minimization problem. As the degree of data heterogeneity increases, FedAvg converges more slowly. Several variants of FedAvg, such as FedProx, FedNova, and SCAFFOLD, tackle the data heterogeneity issue. All these algorithms commonly aggregate the local solutions using the periodic full aggregation scheme.

**Model Discrepancy** – All local SGD-based algorithms allow the clients to independently train their local models within each communication round. The variance of stochastic gradients and heterogeneous data distribution can lead the local models to different directions on parameter space during the local update steps. We formally define such a discrepancy among the models as follows.

$$\sum_{i=1}^m p_i \| \mathbf{u} - \mathbf{x}^i \|^2,$$

where $m$ is the number of clients, $\mathbf{u}$ is the synchronized model, and $\mathbf{x}^i$ is client $i$'s local model. This quantity bounds the difference between the local gradients and the global gradients under a smoothness assumption on objective functions.

## 4   Layer-wise Adaptive Model Aggregation

**Layer Prioritization** – In theoretical analysis, it is common to assume the smoothness of objective functions such that the difference between local gradients and global gradients is bounded by a scaled difference of the corresponding sets of parameters. Motivated by this convention, we define 'layer-wise unit model discrepancy', a useful metric for prioritizing the layers as follows.

$$d_l = \frac{\sum_{i=1}^m p_i \| \mathbf{u}_l - \mathbf{x}_l^i \|^2}{\tau_l(\dim(\mathbf{u}_l))}, \quad l \in \{1, \cdots, L\} \tag{2}$$

where $L$ is the number of layers, $l$ is the layer index, $\mathbf{u}$ is the global parameters, $\mathbf{x}^i$ is the client $i$'s local parameters, $\tau$ is the aggregation interval, and $\dim(\cdot)$ is the number of parameters.

This metric quantifies how much each parameter contributes to the model discrepancy at each iteration. The communication cost is proportional to the number of parameters. Thus, $\sum_{i=1}^m p_i \| \mathbf{u}_l - \mathbf{x}_l^i \|^2 / \dim(\mathbf{u}_l)$ shows how much model discrepancy can be eliminated by synchronizing the layer at a unit communication cost. This metric allows prioritizing the layers such that the layers with a smaller $d_l$ value has a lower priority than the others.

**Adaptive Model Aggregation Algorithm** – We propose FedLAMA, a layer-wise adaptive model aggregation scheme. Algorithm 1 shows FedLAMA algorithm. There are two input parameters: $\tau'$ is the base aggregation interval and $\phi$ is the interval increase factor. First, the parameters at layer $l$ are synchronized across the clients after every $\tau_l$ iterations (line 6). Then, the proposed metric

---

**Algorithm 2:** Layer-wise Adaptive Interval Adjustment.

---

**Input: d**: the observed model discrepancy at all $L$ layers, $\tau'$: the base aggregation interval, $\phi$: the interval increasing factor.

1  Sorted model discrepancy: $\hat{\mathbf{d}} \leftarrow \text{sort}(\mathbf{d})$;

2  Sorted index of the layers: $\hat{\mathbf{i}} \leftarrow \text{argsort}(\mathbf{d})$;

3  Total model size: $\lambda \leftarrow \sum_{l=1}^{L} \dim(\mathbf{u}_l)$;

4  Total model discrepancy: $\delta \leftarrow \sum_{l=1}^{L} d_l * \dim(\mathbf{u}_l)$;

5  **for** $l = 1$ *to* $L$ **do**

6     $\delta_l \leftarrow (\sum_{i=1}^{l} \hat{d}_i * \dim(\mathbf{u}_i))/\delta$;

7     $\lambda_l \leftarrow (\sum_{i=1}^{l} \dim(\mathbf{u}_i))/\lambda$;

8     Find the layer index: $i \leftarrow \hat{i}_l$ ;

9     **if** $\delta_l < \lambda_l$ **then**

10        $\tau_i \leftarrow \phi\tau'$;

11     **else**

12        $\tau_i \leftarrow \tau'$;

13  **Output:** $\tau$: the adjusted aggregation intervals at all $L$ layers.;

---

$d_l$ is calculated using the synchronized parameters $\mathbf{u}_l$ (line 7). At the end of every $\phi\tau'$ iterations, FedLAMA adjusts the model aggregation interval at all the $L$ layers. (line 9).

Algorithm 2 finds the layers that can be less frequently aggregated making a minimal impact on the total model discrepancy. First, the layer-wise degree of model discrepancy is estimated as follows.

$$\delta_l = \frac{\sum_{i=1}^{l} \hat{d}_i * \dim(\mathbf{u}_i)}{\sum_{i=1}^{L} \hat{d}_i * \dim(\mathbf{u}_i)}, \tag{3}$$

where $\hat{d}_i$ is the $i^{th}$ smallest element in the sorted list of the proposed metric $d$. Given $l$ layers with the smallest $d_l$ values, $\delta_l$ quantifies their contribution to the total model discrepancy. Second, the communication cost impact is estimated as follows.

$$\lambda_l = \frac{\sum_{i=1}^{l} \dim(\mathbf{u}_i)}{\sum_{i=1}^{L} \dim(\mathbf{u}_i)} \tag{4}$$

$\lambda_l$ is the ratio of the parameters at the $l$ layers with the smallest $d_l$ values. Thus, $1 - \lambda_l$ estimates the number of parameters that will be more frequently synchronized than the others. As $l$ increases, $\delta_l$ increases while $1 - \lambda_l$ decreases monotonically. Algorithm 2 loops over the $L$ layers finding the $l$ value that makes $\delta_l$ and $1 - \lambda_l$ similar. In this way, it finds the aggregation interval setting that slightly sacrifices the model discrepancy while remarkably reducing the communication cost.

Figure 1 shows the $\delta_l$ and $1 - \lambda_l$ curves collected from a) CIFAR-10 (ResNet20) training and b) CIFAR-100 (Wide-ResNet28-10) training. The x-axis is the number of layers to increase the aggregation interval and the y-axis is the $\delta_l$ and $1 - \lambda_l$ values. The cross point of the two curves is much lower than 0.5 on y-axis in both charts. For instance, in Figure 1.a), the two curves are crossed when $x$ value is 9, and the corresponding $y$ value is near 0.2. That is, when the aggregation interval is increased at those 9 layers, $20\%$ of the total model discrepancy will increase by a factor of $\phi$ while $80\%$ of the total communication cost will decrease by the same factor. Note that the cross points are below 0.5 since the $\delta_l$ and $1 - \lambda_l$ are calculated using the $d_l$ values sorted in an increasing order.

It is worth noting that FedLAMA can be easily extended to improve the convergence rate at the cost of having minor extra communications. In this work, we do not consider finding such interval settings because it can increase the latency cost, which is not desired in Federated Learning. However, in the environments where the latency cost can be ignored, such as high-performance computing platforms, FedLAMA can accelerate the convergence by adjusting the intervals based on the cross point of $1 - \delta_l$ and $\lambda_l$ calculated using the list of $d_l$ values sorted in a decreasing order.

**Impact of Aggregation Interval Increasing Factor** $\phi$ – In Federated Learning, the communication latency cost is usually not negligible, and the total number of communications strongly affects the

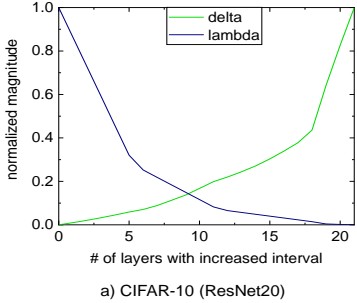 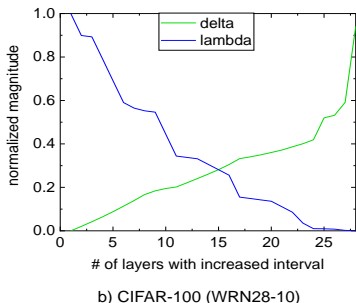

a) CIFAR-10 (ResNet20)  b) CIFAR-100 (WRN28-10)

Figure 1: The comparison between the model discrepancy increase factor $\delta_l$ and the communication cost decrease factor $1 - \lambda_l$ for a) CIFAR-10 and b) CIFAR-100 training.

scalability. When increasing the aggregation interval, Algorithm 2 multiplies a pre-defined small constant $\phi$ to the fixed base interval $\tau'$ (line 10). This approach ensures that the communication latency cost is not increased while the network bandwidth consumption is reduced by a factor of $\phi$.

FedAvg can be considered as a special case of FedLAMA where $\phi$ is set to 1. When $\phi > 1$, FedLAMA less frequently synchronize a subset of layers, and it results in reducing their communication costs. When increasing the aggregation interval, FedLAMA multiplies $\phi$ to the base interval $\tau'$. So, it is guaranteed that the whole model parameters are fully synchronized after $\phi\tau'$ iterations. Because of the layers with the base aggregation interval $\tau'$, the total model discrepancy of FedLAMA after $\phi\tau'$ iterations is always smaller than that of FedAvg with an aggregation interval of $\phi\tau'$.

## 5 CONVERGENCE ANALYSIS

### 5.1 PRELIMINARIES

**Notations** – All vectors in this paper are column vectors. $\mathbf{x} \in \mathbb{R}^d$ denotes the parameters of one local model and $m$ is the number of clients. The stochastic gradient computed from a single training data point $\boldsymbol{\xi}$ is denoted by $g(\mathbf{x}, \boldsymbol{\xi})$. For convenience, we use $g(\mathbf{x})$ instead. The full batch gradient is denoted by $\nabla F(\mathbf{x})$. We use $\|\cdot\|$ and $\|\cdot\|_{op}$ to denote $l2$ norm and matrix operator norm, respectively.

**Assumptions** – We analyze the convergence rate of FedLAMA under the following assumptions.

1. (Smoothness). Each local objective function is L-smooth, that is, $\|\nabla F_i(\mathbf{x}) - \nabla F_i(\mathbf{y})\| \leq L\|\mathbf{x} - \mathbf{y}\|, \forall i \in \{1, \cdots, m\}$.
2. (Unbiased Gradient). The stochastic gradient at each client is an unbiased estimator of the local full-batch gradient: $\mathbb{E}_\xi[g_i(\mathbf{x}, \xi)] = \nabla F_i(\mathbf{x})$.
3. (Bounded Variance). The stochastic gradient at each client has bounded variance: $\mathbb{E}_\xi[\|g_i(\mathbf{x}, \xi) - \nabla F_i(\mathbf{x})\|^2 \leq \sigma^2], \forall i \in \{1, \cdots, m\}, \sigma^2 \geq 0$.
4. (Bounded Dissimilarity). For any sets of weights $\{p_i \geq 0\}_{i=1}^m, \sum_{i=1}^m p_i = 1$, there exist constants $\beta^2 \geq 1$ and $\kappa^2 \geq 0$ such that $\sum_{i=1}^m p_i\|\nabla F_i(\mathbf{x})\|^2 \leq \beta^2\|\sum_{i=1}^m p_i \nabla F_i(\mathbf{x})\|^2 + \kappa^2$. If local objective functions are identical to each other, $\beta^2 = 1$ and $\kappa^2 = 0$.

### 5.2 ANALYSIS

We begin with showing two key lemmas. All the proofs can be found in Appendix.

**Lemma 5.1.** *(Framework) Under Assumption* $1 \sim 3$*, if the learning rate satisfies* $\eta \leq \frac{1}{2L}$*, FedLAMA ensures*

$$\frac{1}{K}\sum_{k=1}^K \mathbb{E}\left[\|\nabla F(\mathbf{u}_k)\|^2\right] \leq \frac{2}{\eta K}\mathbb{E}\left[F(\mathbf{u}_1) - F(\mathbf{u}_*)\right] + 2\eta L\sigma^2 \sum_{i=1}^m (p_i)^2$$

$$+ \frac{L^2}{K}\sum_{k=1}^K\sum_{i=1}^m p_i \mathbb{E}\left[\|\mathbf{u}_k - \mathbf{x}_k^i\|^2\right]. \tag{5}$$

**Lemma 5.2.** *(Model Discrepancy) Under Assumption* $1 \sim 4$*, if the learning rate satisfies* $\eta < \frac{1}{2(\tau_{max}-1)L}$*, FedLAMA ensures*

$$
\frac{1}{K} \sum_{k=1}^{K} \sum_{i=1}^{m} p_i \, \mathbb{E} \left[ \left\| \mathbf{u}_k - \mathbf{x}_k^i \right\|^2 \right] \leq \frac{2\eta^2(\tau_{max}-1)\sigma^2}{1-A} + \frac{A\kappa^2}{L^2(1-A)}
$$
$$
+ \frac{A\beta^2}{KL^2(1-A)} \sum_{k=1}^{K} \mathbb{E} \left[ \left\| \nabla F(\mathbf{u}_k) \right\|^2 \right],
$$

(6)

*where* $A = 4\eta^2(\tau_{max}-1)^2 L^2$ *and* $\tau_{max}$ *is the largest averaging interval across all the layers.*

Based on Lemma 5.1 and 5.2, we analyze the convergence rate of FedLAMA as follows.

**Theorem 5.3.** *Suppose all* $m$ *local models are initialized to the same point* $\mathbf{u}_1$*. Under Assumption* $1 \sim 4$*, if FedLAMA runs for* $K$ *iterations and the learning rate satisfies* $\eta \leq \min \left\{ \frac{1}{2(\tau_{max}-1)L}, \frac{1}{L\sqrt{2\tau_{max}(\tau_{max}-1)(2\beta^2+1)}} \right\}$*, FedLAMA ensures*

$$
\mathbb{E} \left[ \frac{1}{K} \sum_{i=1}^{K} \left\| \nabla F(\mathbf{u}_k) \right\|^2 \right] \leq \frac{4}{\eta K} \left( \mathbb{E} \left[ F(\mathbf{u}_1) - F(\mathbf{u}_*) \right] \right) + 4\eta \sum_{i=1}^{m} p_i^2 L \sigma^2
$$

(7)

$$
+ 3\eta^2(\tau_{max}-1)L^2\sigma^2 + 6\eta^2 \tau_{max}(\tau_{max}-1)L^2\kappa^2,
$$

*where* $\mathbf{u}_*$ *indicates a local minimum and* $\tau_{max}$ *is the largest averaging interval across all the layers.*

**Remark 1.** (Linear Speedup) With a sufficiently small diminishing learning rate and a large number of training iterations, FedLAMA achieves linear speedup. If the learning rate is $\eta = \frac{\sqrt{m}}{\sqrt{K}}$ and $p_i = \frac{1}{m}, \forall i \in \{1, \cdots, m\}$, we have

$$
\mathbb{E} \left[ \frac{1}{K} \sum_{i=1}^{K} \left\| \nabla F(\mathbf{u}_k) \right\|^2 \right] \leq \mathcal{O} \left( \frac{1}{\sqrt{mK}} \right) + \mathcal{O} \left( \frac{m}{K} \right)
$$

(8)

If $K > m^3$, the first term on the right-hand side becomes dominant and it achieves linear speedup.

**Remark 2.** (Impact of Interval Increase Factor $\phi$) The worst-case model discrepancy depends on the largest averaging interval across all the layers, $\tau_{max} = \phi\tau'$. The larger the interval increase factor $\phi$, the larger the model discrepancy terms in (7). In the meantime, as $\phi$ increases, the communication frequency at the selected layers is proportionally reduced. So, $\phi$ should be appropriately tuned to effectively reduce the communication cost while not much increasing the model discrepancy.

## 6 EXPERIMENTS

**Experimental Settings** – We evaluate FedLAMA using three representative benchmark datasets: CIFAR-10 (ResNet20 (He et al. (2016))), CIFAR-100 (WideResNet28-10 (Zagoruyko & Komodakis (2016))), and Federated Extended MNIST (CNN (Caldas et al. (2018))). We use TensorFlow 2.4.3 for local training and MPI for model aggregation. All our experiments are conducted on 4 compute nodes each of which has 2 NVIDIA v100 GPUs.

Due to the limited compute resources, we simulate Federated Learning such that each process sequentially trains multiple models and then the models are aggregated across all the processes at once. While it provides the same classification results as the actual Federated Learning, the training time is serialized within each process. Thus, instead of wall-clock time, we consider the total communication cost calculated as follows.

$$
\mathcal{C} = \sum_{l=1}^{L} \mathcal{C}_l = \sum_{l=1}^{L} \dim(\mathbf{u}_l) * \kappa_l,
$$

(9)

where $\kappa_l$ is the total number of communications at layer $l$ during the training.

Table 1: (IID data) CIFAR-10 classification results. The number of workers is 128 and the local batch size is 32 in all the experiments. The epoch budget is 300.

| LR | Base aggregation interval: $\tau'$ | Interval increase factor: $\phi$ | Validation acc. | Comm. cost |
|---|---|---|---|---|
| 0.8 | 6 | 1 (FedAvg) | $88.37 \pm 0.02\%$ | 100% |
| 0.8 | 12 | 1 (FedAvg) | $84.74 \pm 0.05\%$ | 50% |
| 0.4 | 6 | 2 (FedLAMA) | **88.41** $\pm 0.01\%$ | **62.33**% |
| 0.6 | 24 | 1 (FedAvg) | $80.34 \pm 0.3\%$ | 25% |
| 0.6 | 6 | 4 (FedLAMA) | **86.21** $\pm 0.1\%$ | **42.17**% |

Table 2: (IID data) CIFAR-100 classification results. The number of workers is 128 and the local batch size is 32 in all the experiments. The epoch budget is 250.

| LR | Base aggregation interval: $\tau'$ | Interval increase factor: $\phi$ | Validation acc. | Comm. cost |
|---|---|---|---|---|
| 0.6 | 6 | 1 (FedAvg) | $76.50 \pm 0.02\%$ | 100% |
| 0.6 | 12 | 1 (FedAvg) | $66.97 \pm 0.9\%$ | 50% |
| 0.5 | 6 | 2 (FedLAMA) | **76.02** $\pm 0.01\%$ | **66.01**% |
| 0.6 | 24 | 1 (FedAvg) | $45.01 \pm 1.1\%$ | 25% |
| 0.5 | 6 | 4 (FedLAMA) | **76.17** $\pm 0.02\%$ | **39.91**% |

**Hyper-Parameter Settings** – We use 128 clients in our experiments. The local batch size is set to 32 and the learning rate is tuned based on a grid search. For CIFAR-10 and CIFAR-100, we artificially generate heterogeneous data distributions using Dirichlet's distribution. When using Non-IID data, we also consider partial device participation such that randomly chosen 25% of the clients participate in training at every $\phi\tau'$ iterations. We report the average accuracy across at least three separate runs.

## 6.1 CLASSIFICATION PERFORMANCE ANALYSIS

To evaluate the proposed model aggregation scheme, we keep all the other factors the same, such as optimizer, the number of clients, the degree of heterogeneity, and compare the performance across different model aggregation schemes. We compare the performance across three different model aggregation settings as follows.

- Periodic full aggregation with an interval of $\tau'$
- Periodic full aggregation with an interval of $\phi\tau'$
- Layer-wise adaptive aggregation with intervals of $\tau'$ and $\phi$

The first setting provides the baseline communication cost, and we compare it to the other settings' communication costs. The third setting is FedLAMA with the base aggregation interval $\tau'$ and the interval increase factor $\phi$. Due to the limited space, we present a part of experimental results that deliver the key insights. More results can be found in Appendix.

**Experimental Results with IID Data** – We first present CIFAR-10 and CIFAR-100 classification results under IID data settings. Table 1 and 2 show the CIFAR-10 and CIFAR-100 results, respectively. Note that the learning rate is individually tuned for each setting using a grid search, and we report the best settings. In both tables, the first row shows the performance of FedAvg with a short interval $\tau' = 6$. As the interval increases, FedAvg significantly loses the accuracy while the communication cost is proportionally reduced. FedLAMA achieves a comparable accuracy to FedAvg with $\tau' = 6$ while its communication cost is similar to that of FedAvg with $\phi\tau'$. These results demonstrate that Algorithm 2 effectively finds the layer-wise interval settings that maximize the communication cost reduction while minimizing the model discrepancy increase.

**Experimental Results with Non-IID Data** – We now evaluate the performance of FedLAMA using non-IID data. FEMNIST is inherently heterogeneous such that it contains the hand-written digit pictures collected from 3, 550 different writers. We use random 10% of the writers' training samples in our experiments. Table 3 shows the FEMNIST classification results. The base interval $\tau'$ is set to 10. FedAvg ($\phi = 1$) significantly loses the accuracy as the aggregation interval increases. For example, when the interval increases from 10 to 40, the accuracy is dropped by $2.1\% \sim 2.7\%$. In contrast, FedLAMA maintains the accuracy when $\phi$ increases, while the communication cost is remarkably reduced. This result demonstrates that FedLAMA effectively finds the best interval setting that reduces the communication cost while maintaining the accuracy.

Table 3: (Non-IID data) FEMNIST classification results. The number of workers is 128 and the local batch size is 32 in all the experiments. The number of training iterations is $2,000$.

| LR | Base aggregation interval: $\tau'$ | Interval increase factor: $\phi$ | active ratio | Validation acc. | Comm. cost |
|---|---|---|---|---|---|
| | 10 | 1 (FedAvg) | | $86.04 \pm 0.01\%$ | $100\%$ |
| | 20 | 1 (FedAvg) | | $85.38 \pm 0.02\%$ | $50\%$ |
| 0.04 | 10 | 2 (FedLAMA) | $25\%$ | $\mathbf{86.01} \pm 0.01\%$ | $\mathbf{52.83}\%$ |
| | 40 | 1 (FedAvg) | | $83.97 \pm 0.02\%$ | $25\%$ |
| | 10 | 4 (FedLAMA) | | $\mathbf{85.61} \pm 0.02\%$ | $\mathbf{29.97}\%$ |
| | 10 | 1 (FedAvg) | | $86.59 \pm 0.01\%$ | $100\%$ |
| | 20 | 1 (FedAvg) | | $85.50 \pm 0.02\%$ | $50\%$ |
| 0.04 | 10 | 2 (FedLAMA) | $50\%$ | $\mathbf{86.07} \pm 0.02\%$ | $\mathbf{53.32}\%$ |
| | 40 | 1 (FedAvg) | | $83.92 \pm 0.02\%$ | $25\%$ |
| | 10 | 4 (FedLAMA) | | $\mathbf{85.77} \pm 0.02\%$ | $\mathbf{29.98}\%$ |
| | 10 | 1 (FedAvg) | | $85.74 \pm 0.03\%$ | $100\%$ |
| | 20 | 1 (FedAvg) | | $85.08 \pm 0.01\%$ | $50\%$ |
| 0.04 | 10 | 2 (FedLAMA) | $100\%$ | $\mathbf{85.40} \pm 0.02\%$ | $\mathbf{51.86}\%$ |
| | 40 | 1 (FedAvg) | | $83.62 \pm 0.02\%$ | $25\%$ |
| | 10 | 4 (FedLAMA) | | $\mathbf{84.67} \pm 0.02\%$ | $\mathbf{29.98}\%$ |

Table 4: (Non-IID data) CIFAR-10 classification results. The number of workers is 128 and the local batch size is 32 in all the experiments. The number of training iterations is $6,000$.

| LR | Base aggregation interval: $\tau'$ | Interval increase factor: $\phi$ | active ratio | Dirichlet's coeff. | Validation acc. | Comm. cost |
|---|---|---|---|---|---|---|
| | 6 | 1 (FedAvg) | | | $84.02 \pm 0.1\%$ | $100\%$ |
| 0.4 | 24 | 1 (FedAvg) | $25\%$ | 0.1 | $76.27 \pm 0.08\%$ | $25\%$ |
| | 6 | 4 (FedLAMA) | | | $\mathbf{83.06} \pm 0.1\%$ | $\mathbf{39.52}\%$ |
| | 6 | 1 (FedAvg) | | | $87.59 \pm 0.2\%$ | $100\%$ |
| 0.8 | 24 | 1 (FedAvg) | $25\%$ | 0.5 | $83.36 \pm 0.4\%$ | $25\%$ |
| | 6 | 4 (FedLAMA) | | | $\mathbf{86.57} \pm 0.02\%$ | $\mathbf{42.40}\%$ |
| | 6 | 1 (FedAvg) | | | $89.52 \pm 0.05\%$ | $100\%$ |
| 0.8 | 24 | 1 (FedAvg) | $100\%$ | 0.1 | $84.82 \pm 0.06\%$ | $25\%$ |
| | 6 | 4 (FedLAMA) | | | $\mathbf{87.47} \pm 0.1\%$ | $\mathbf{42.49}\%$ |
| | 6 | 1 (FedAvg) | | | $90.53 \pm 0.08\%$ | $100\%$ |
| 0.8 | 24 | 1 (FedAvg) | $100\%$ | 0.5 | $85.68 \pm 0.1\%$ | $25\%$ |
| | 6 | 4 (FedLAMA) | | | $\mathbf{87.45} \pm 0.05\%$ | $\mathbf{42.73}\%$ |

Table 5: (Non-IID data) CIFAR-100 classification results. The number of workers is 128 and the local batch size is 32 in all the experiments. The number of training iterations is $6,000$.

| LR | Base aggregation interval: $\tau'$ | Interval increase factor: $\phi$ | active ratio | Dirichlet's coeff. | Validation acc. | Comm. cost |
|---|---|---|---|---|---|---|
| | 6 | 1 (FedAvg) | | | $79.15 \pm 0.02\%$ | $100\%$ |
| 0.4 | 12 | 1 (FedAvg) | $25\%$ | 0.1 | $76.16 \pm 0.05\%$ | $50\%$ |
| | 6 | 2 (FedLAMA) | | | $\mathbf{78.63} \pm 0.03\%$ | $\mathbf{63.14}\%$ |
| | 6 | 1 (FedAvg) | | | $78.81 \pm 0.1\%$ | $100\%$ |
| 0.4 | 12 | 1 (FedAvg) | $25\%$ | 0.5 | $76.11 \pm 0.05\%$ | $50\%$ |
| | 6 | 2 (FedLAMA) | | | $\mathbf{77.86} \pm 0.04\%$ | $\mathbf{63.20}\%$ |
| | 6 | 1 (FedAvg) | | | $79.77 \pm 0.04\%$ | $100\%$ |
| 0.4 | 12 | 1 (FedAvg) | $100\%$ | 0.1 | $77.71 \pm 0.08\%$ | $50\%$ |
| | 6 | 2 (FedLAMA) | | | $\mathbf{79.07} \pm 0.1\%$ | $\mathbf{60.48}\%$ |
| | 6 | 1 (FedAvg) | | | $80.19 \pm 0.05\%$ | $100\%$ |
| 0.4 | 12 | 1 (FedAvg) | $100\%$ | 0.5 | $77.40 \pm 0.06\%$ | $50\%$ |
| | 6 | 2 (FedLAMA) | | | $\mathbf{78.88} \pm 0.05\%$ | $\mathbf{61.73}\%$ |

Table 4 and 5 show the non-IID CIFAR-10 and CIFAR-100 experimental results. We use Dirichlet's distribution to generate heterogeneous data across all the clients. The detailed settings regarding Dirichlet's distribution can be found in Appendix. The base aggregation interval $\tau'$ is set to 6. The interval increase factor $\phi$ is set to 2 for FedLAMA. Likely to the IID data experiments, we observe that the periodic full averaging significantly loses the accuracy as the model aggregation interval increases, while it has a proportionally reduced communication cost. For both datasets, FedLAMA achieves a comparable accuracy to the periodic full averaging with the interval of $\tau'$ while having the communication cost that is close to the periodic full averaging with the increased interval of $\phi\tau'$. Especially, FedLAMA works effectively even when the Dirichlet's coefficient is set to $0.1$. The coefficient of $0.1$ represents an extremely high degree of data heterogeneity in terms of not only the number of samples per client but also the balance of the classes assigned to each client. These results imply that FedLAMA is a practical algorithm for Federated Learning applications with highly heterogeneous data distributions.

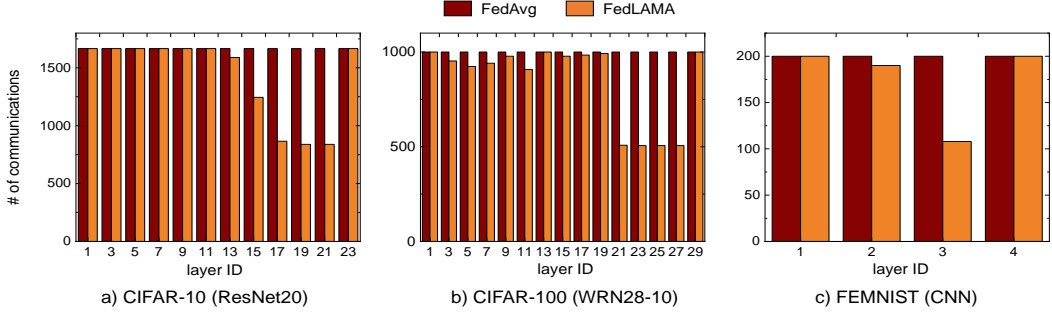

Figure 2: The number of communications at the individual layers. The communications are counted during the whole training (non-IID data).

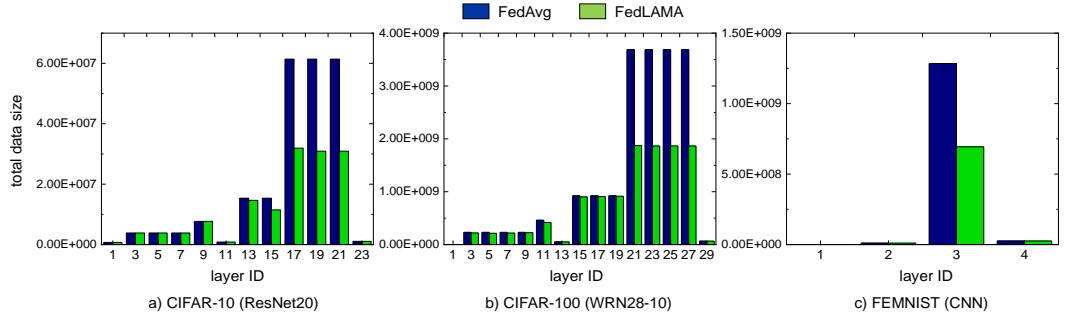

Figure 3: The total data size (communication cost) that correspond to Figure 2. The data size comparison clearly shows where the performance gain of FedLAMA comes from.

## 6.2 COMMUNICATION EFFICIENCY ANALYSIS

We analyze the total number of communications and the accumulated data size to evaluate the communication efficiency of FedLAMA. Figure 2 shows the total number of communications at the individual layers. The $\tau'$ is set to 6 and $\phi$ is 2 for FedLAMA. The key insight is that FedLAMA increases the aggregation interval mostly at the output-side large layers. This means the $d_l$ value shown in Equation (2) at the these layers are smaller than the others. Since these large layers take up most of the total model parameters, the communication cost is remarkably reduced when their aggregation intervals are increased. Figure 3 shows the layer-wise local data size shown in Equation 9. FedLAMA shows the significantly smaller total data size than FedAvg. The extra computational cost of FedLAMA is almost negligible since it calculates $d_l$ after each communication round only. Therefore, given the virtually same computational cost, FedLAMA aggregates the local models at a cheaper communication cost, and thus it improves the scalablity of Federated Learning.

We found that the amount of the reduced communication cost was not strongly affected by the degree of data heterogeneity. As shown in Table 4 and 5, the reduced communication cost is similar across different Dirichlet's coefficients and device participation ratios. That is, FedLAMA can be considered as an effective model aggregation scheme regardless of the degree of data heterogeneity.

## 7 CONCLUSION

We proposed a layer-wise model aggregation scheme that adaptively adjusts the model aggregation interval at run-time. Breaking the convention of aggregating the whole model parameters at once, this novel model aggregation scheme introduces a flexible communication strategy for scalable Federated Learning. Furthermore, we provide a solid convergence guarantee of FedLAMA under the assumptions on the non-convex objective functions and the non-IID data distribution. Our empirical study also demonstrates the efficacy of FedLAMA for scalable and accurate Federated Learning.

Harmonizing FedLAMA with other advanced optimizers, gradient compression, and low-rank approximation methods is a promising future work.

## 8 CODE OF ETHICS

Our work does not deliver potentially harmful insights or conflicts of interests. We also do not find any potential inappropriate application or privacy/security issues. The datasets we used in our study are all public benchmark datasets, and our source code will be opened once the paper is accepted.

## 9 REPRODUCIBILITY STATEMENT

The software versions, implementation details, hyper-parameter settings can be found in the first two paragraphs of Section 6. The entire source code used in our experiments will be published as an open source once the paper is accepted. We believe one can exactly reproduce our experimental results following the provided descriptions.

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

# A APPENDIX

## A.1 CONVERGENCE ANALYSIS

Herein, we provide the proofs of the lemmas and theorem shown in Section 5.

### A.1.1 PRELIMINARIES

FedLAMA periodically chooses a few layers that will be less frequently synchronized. We call these layers Least Critical Layers (LCL) for short.

**Notations** – All vectors in this paper are column vectors. $\mathbf{x} \in \mathbb{R}^d$ denotes the parameters of one local model and $m$ is the number of workers. The stochastic gradient computed from a single training data point $\boldsymbol{\xi}$ is denoted by $g(\mathbf{x}, \boldsymbol{\xi})$. For convenience, we use $g(\mathbf{x})$ instead. The full batch gradient is denoted by $\nabla F(\mathbf{x})$. We use $\| \cdot \|$ and $\| \cdot \|_{op}$ to denote $l2$ norm and matrix operator norm, respectively.

**Objective Function** – In this paper, we consider federated optimization problems as follows.

$$\min_{\mathbf{x} \in \mathbb{R}^d} \left[ F(\mathbf{x}) := \sum_{i=1}^{m} p_i F_i(\mathbf{x}) \right], \tag{10}$$

where $p_i = n_i/n$ is the ratio of local data to the total dataset, and $F_i(\mathbf{x}) = \frac{1}{n_i} \sum_{\xi \in \mathcal{D}} f_i(\mathbf{x}, \xi)$ is the local objective function of client $i$. $n$ is the global dataset size and $n_i$ is the local dataset size. Note that, by definition, $\sum_{i=1}^{m} p_i = 1$.

**Averaging Matrix** – We define a time-varying averaging matrix $\mathbf{W}_k \in \mathbb{R}^{dm \times dm}$ as follows.

$$\mathbf{W_k} = \begin{cases} \mathbf{P}, & \text{if } k \bmod \tau_{min} \text{ is } 0 \\ \mathbf{J}, & \text{if } k \bmod \tau_{max} \text{ is } 0 \\ \mathbf{I}, & \text{otherwise} \end{cases} \tag{11}$$

$\mathbf{I}$ is an identity matrix, $\mathbf{P}$ is also a time-varying averaging matrix, and $\mathbf{J}$ is a full averaging matrix. First, $\mathbf{P}_i^1$ is a $d \times d$ diagonal matrix that has 1 for the diagonal elements that correspond to the LCL parameters and $p_i$ for all the other diagonal elements. Likewise, $\mathbf{P}_i^0$ is another $d \times d$ diagonal matrix that has 0 for the diagonal elements that correspond to the LCL parameters and $p_i$ for all the other diagonal elements. Then, $\mathbf{P}$ is defined as follows.

$$\mathbf{P} = \begin{cases} \mathbf{P}^1, & \text{for } m \text{ diagonal blocks} \\ \mathbf{P}^0, & \text{for all the other blocks} \end{cases} \tag{12}$$

The $i^{th}$ block column of $\mathbf{P}$ consists of $\mathbf{P}_i^1$ and $\mathbf{P}_i^0$ following the above definition.

Here we present an example of $\mathbf{P}$ where $m = 2$ and $d = 2$. In this example, $p_0 = 1/3$ and $p_1 = 2/3$. Saying the LCL is the second parameter, $\mathbf{P}$ is defined as follows.

$$\mathbf{P}_0^1 = \begin{bmatrix} \frac{1}{3} & 0 \\ 0 & 1 \end{bmatrix}, \mathbf{P}_0^0 = \begin{bmatrix} \frac{1}{3} & 0 \\ 0 & 0 \end{bmatrix}, \mathbf{P}_1^1 = \begin{bmatrix} \frac{2}{3} & 0 \\ 0 & 1 \end{bmatrix}, \mathbf{P}_1^0 = \begin{bmatrix} \frac{2}{3} & 0 \\ 0 & 0 \end{bmatrix} \tag{13}$$

$$\mathbf{P} = \begin{bmatrix} \mathbf{P}_0^1 & \mathbf{P}_1^0 \\ \mathbf{P}_0^0 & \mathbf{P}_1^1 \end{bmatrix} = \begin{bmatrix} \frac{1}{3} & 0 & \frac{2}{3} & 0 \\ 0 & 1 & 0 & 0 \\ \frac{1}{3} & 0 & \frac{2}{3} & 0 \\ 0 & 0 & 0 & 1 \end{bmatrix}. \tag{14}$$

The full-averaging matrix $\mathbf{J}$ is defined as follows. First, $\mathbf{J}_i$ is a $d \times d$ diagonal matrix that has $p_i$ for the diagonal elements. Then, $\mathbf{J}$ consists of $m \times m$ blocks of $\mathbf{J}_i$ such that each column block is $m$ of $\mathbf{J}_i$ blocks. Here we present an example of $\mathbf{J}$ where $m = 2$ and $d = 2$ as follows.

$$\mathbf{J}_0 = \begin{bmatrix} \frac{1}{3} & 0 \\ 0 & \frac{1}{3} \end{bmatrix}, \mathbf{J}_1 = \begin{bmatrix} \frac{2}{3} & 0 \\ 0 & \frac{2}{3} \end{bmatrix} \tag{15}$$

$$\mathbf{J} = \begin{bmatrix} \mathbf{J}_0 & \mathbf{J}_1 \\ \mathbf{J}_0 & \mathbf{J}_1 \end{bmatrix} = \begin{bmatrix} \frac{1}{3} & 0 & \frac{2}{3} & 0 \\ 0 & \frac{1}{3} & 0 & \frac{2}{3} \\ \frac{1}{3} & 0 & \frac{2}{3} & 0 \\ 0 & \frac{1}{3} & 0 & \frac{2}{3} \end{bmatrix}. \tag{16}$$

The averaging matrix $\mathbf{P}$ and $\mathbf{J}$ have the following properties:

1. $\mathbf{P1}_{dm} = \mathbf{1}_{dm}, \mathbf{J1}_{dm} = \mathbf{1}_{dm}$.
2. The product of any two averaging matrices consists only of diagonal block matrices because all the blocks in $\mathbf{P}$ and $\mathbf{J}$ are diagonal.
3. $\mathbf{PJ} = \mathbf{JP} = \mathbf{J}$ regardless of which layers are chosen as the LCL.
4. $\mathbf{PP} = \mathbf{P}$ regardless of which layers are chosen as the LCL.

**Vectorization** – We define a vectorized form of $m$ local model parameters $\mathbf{x}_k \in \mathbb{R}^{dm}$, its stochastic gradients $\mathbf{g}_k \in \mathbb{R}^{dm}$, and the full gradients $\mathbf{f}_k \in \mathbb{R}^{dm}$ as follows

$$\begin{aligned}
\mathbf{x}_k &= vec\left\{\mathbf{x}_k^1, \mathbf{x}_k^2, \cdots, \mathbf{x}_k^m\right\} \\
\mathbf{g}_k &= vec\left\{g_1(\mathbf{x}_k^1), g_2(\mathbf{x}_k^2), \cdots, g_m(\mathbf{x}_k^m)\right\} \\
\mathbf{f}_k &= vec\left\{\nabla F_1(\mathbf{x}_k^1), \nabla F_2(\mathbf{x}_k^2), \cdots, \nabla F_m(\mathbf{x}_k^m)\right\}.
\end{aligned} \tag{17}$$

The full model aggregation can be written using the vectorized form of local models $\mathbf{x}_k$ and the averaging matrix $\mathbf{J}$ as follows.

$$\mathbf{Jx}_k = \begin{bmatrix} \frac{1}{3} & 0 & \frac{2}{3} & 0 \\ 0 & \frac{1}{3} & 0 & \frac{2}{3} \\ \frac{1}{3} & 0 & \frac{2}{3} & 0 \\ 0 & \frac{1}{3} & 0 & \frac{2}{3} \end{bmatrix} \begin{bmatrix} x_k^{(1,1)} \\ x_k^{(1,2)} \\ x_k^{(2,1)} \\ x_k^{(2,2)} \end{bmatrix} = \begin{bmatrix} (x_k^{(1,1)} + 2x_k^{(2,1)})/3 \\ (x_k^{(1,2)} + 2x_k^{(2,2)})/3 \\ (x_k^{(1,1)} + 2x_k^{(2,1)})/3 \\ (x_k^{(1,2)} + 2x_k^{(2,2)})/3 \end{bmatrix} \tag{18}$$

where $x_k^{(i,j)}$ is the $j^{th}$ model parameter of local model $i$ at iteration $k$.

We also define the following additional vectorized forms of the weighted model parameters and gradients for convenience.

$$\begin{aligned}
\hat{\mathbf{x}}_k &= vec\left\{\sqrt{p_1}\mathbf{x}_k^1, \sqrt{p_2}\mathbf{x}_k^2, \cdots, \sqrt{p_m}\mathbf{x}_k^m\right\} \\
\hat{\mathbf{g}}_k &= vec\left\{\sqrt{p_1}g_1(\mathbf{x}_k^1), \sqrt{p_2}g_2(\mathbf{x}_k^2), \cdots, \sqrt{p_m}g_m(\mathbf{x}_k^m)\right\} \\
\hat{\mathbf{f}}_k &= vec\left\{\sqrt{p_1}\nabla F_1(\mathbf{x}_k^1), \sqrt{p_2}\nabla F_2(\mathbf{x}_k^2), \cdots, \sqrt{p_m}\nabla F_m(\mathbf{x}_k^m)\right\}
\end{aligned} \tag{19}$$

**Assumptions** – We analyze the convergence rate of FedLAMA under the following assumptions.

1. (Smoothness). Each local objective function is L-smooth, that is, $\|\nabla F_i(\mathbf{x}) - \nabla F_i(\mathbf{y})\| \le L\|\mathbf{x} - \mathbf{y}\|, \forall i \in \{1, \cdots, m\}$.
2. (Unbiased Gradient). The stochastic gradient at each client is an unbiased estimator of the local full-batch gradient: $\mathbb{E}_\xi[g_i(\mathbf{x}, \xi)] = \nabla F_i(\mathbf{x})$.
3. (Bounded Variance). The stochastic gradient at each client has bounded variance: $\mathbb{E}_\xi[\|g_i(\mathbf{x}, \xi) - \nabla F_i(\mathbf{x})\|^2 \le \sigma^2], \forall i \in \{1, \cdots, m\}, \sigma^2 \ge 0$.
4. (Bounded Dissimilarity). For any sets of weights $\{p_i \ge 0\}_{i=1}^m, \sum_{i=1}^m p_i = 1$, there exist constants $\beta^2 \ge 1$ and $\kappa^2 \ge 0$ such that $\sum_{i=1}^m p_i\|\nabla F_i(\mathbf{x})\|^2 \le \beta^2\|\sum_{i=1}^m p_i\nabla F_i(\mathbf{x})\|^2 + \kappa^2$. If local objective functions are identical to each other, $\beta^2 = 1$ and $\kappa^2 = 0$.

### A.1.2 PROOFS

**Theorem 5.1.** *Suppose all $m$ local models are initialized to the same point $\mathbf{u}_1$. Under Assumption $1 \sim 4$, if FedLAMA runs for $K$ iterations and the learning rate satisfies $\eta \leq \min\left\{\frac{1}{2(\tau_{max}-1)L}, \frac{1}{L\sqrt{2\tau_{max}(\tau_{max}-1)(2\beta^2+1)}}\right\}$, FedLAMA ensures*

$$\mathbb{E}\left[\frac{1}{K}\sum_{i=1}^{K}\|\nabla F(\mathbf{u}_k)\|^2\right] \leq \frac{4}{\eta K}\left(\mathbb{E}\left[F(\mathbf{u}_1) - F(\mathbf{u}_*)\right]\right) + 4\eta \sum_{i=1}^{m} p_i^2 L \sigma^2$$
$$+ 3\eta^2(\tau_{max}-1)L^2\sigma^2 + 6\eta^2\tau_{max}(\tau_{max}-1)L^2\kappa^2, \quad (20)$$

*where $\mathbf{u}_*$ indicates a local minimum.*

*Proof.* Based on Lemma 5.1 and 5.2, we have

$$\frac{1}{K}\sum_{k=1}^{K}\mathbb{E}\left[\|\nabla F(\mathbf{u}_k)\|^2\right] \leq \frac{2}{\eta K}\left(\mathbb{E}\left[F(\mathbf{u}_1) - F(\mathbf{u}_*)\right]\right) + 2\eta \sum_{i=1}^{m} p_i^2 L \sigma^2$$
$$+ L^2\left(\frac{\eta^2(\tau_{max}-1)\sigma_j^2}{1-A} + \frac{A\beta^2}{KL^2(1-A)}\sum_{k=1}^{K}\mathbb{E}\left[\|\nabla F(\mathbf{u}_k)\|^2\right] + \frac{A\kappa^2}{L^2(1-A)}\right).$$

After re-writing the left-hand side and a minor rearrangement, we have

$$\frac{1}{K}\sum_{k=1}^{K}\mathbb{E}\left[\|\nabla F(\mathbf{u}_k)\|^2\right] \leq \frac{2}{\eta K}\left(\mathbb{E}\left[F(\mathbf{u}_1) - F(\mathbf{u}_*)\right]\right) + 2\eta \sum_{i=1}^{m} p_i^2 L \sigma^2$$
$$+ \frac{1}{K}\sum_{k=1}^{K}\frac{A\beta^2}{1-A}\mathbb{E}\left[\|\nabla F(\mathbf{u}_k)\|^2\right]$$
$$+ L^2\left(\frac{\eta^2(\tau_{max}-1)\sigma^2}{1-A} + \frac{A\kappa^2}{L^2(1-A)}\right).$$

By moving the third term on the right-hand side to the left-hand side, we have

$$\frac{1}{K}\sum_{k=1}^{K}\left(1 - \frac{A\beta^2}{1-A}\right)\mathbb{E}\left[\|\nabla_j F(\mathbf{u}_k)\|^2\right] \leq \frac{2}{\eta K}\left(\mathbb{E}\left[F(\mathbf{u}_1) - F(\mathbf{u}_*)\right]\right) + 2\eta \sum_{i=1}^{m} p_i^2 L \sigma^2$$
$$+ L^2\left(\frac{\eta^2(\tau_{max}-1)\sigma^2}{1-A} + \frac{A\kappa^2}{L^2(1-A)}\right). \quad (21)$$

If $A \leq \frac{1}{2\beta^2+1}$, then $\frac{A\beta^2}{1-A} \leq \frac{1}{2}$. Therefore, (21) can be simplified as follows.

$$\frac{1}{K}\sum_{k=1}^{K}\mathbb{E}\left[\|\nabla F(\mathbf{u}_k)\|^2\right] \leq \frac{4}{\eta K}\left(\mathbb{E}\left[F(\mathbf{u}_1) - F(\mathbf{u}_*)\right]\right) + 4\eta \sum_{i=1}^{m} p_i^2 L \sigma^2 \quad (22)$$
$$+ 2L^2\left(\frac{\eta^2(\tau_{max}-1)\sigma^2}{1-A}\right) + 2\frac{A\kappa^2}{1-A}.$$

The learning rate condition $A \leq \frac{1}{2\beta^2+1}$ also ensures that $\frac{1}{1-A} \leq 1 + \frac{1}{2\beta^2}$. Based on Assumption 4, $\frac{1}{2\beta^2} \leq \frac{2}{3}$, and thus $\frac{1}{1-A} \leq \frac{2}{3}$. Therefore, we have

$$\frac{1}{K}\sum_{k=1}^{K}\mathbb{E}\left[\|\nabla F(\mathbf{u}_k)\|^2\right] \leq \frac{4}{\eta K}\left(\mathbb{E}\left[F(\mathbf{u}_1) - F(\mathbf{u}_*)\right]\right) + 4\eta \sum_{i=1}^{m} p_i^2 L \sigma^2$$
$$+ 3\eta^2(\tau_{max}-1)L^2\sigma^2 + 6\eta^2\tau_{max}(\tau_{max}-1)L^2\kappa^2.$$

We complete the proof. $\square$

**Learning Rate Constraints** – In Theorem 5.3, we have two learning rate constraints, one from (22) and the other from (51) as follows.

$$A < \frac{1}{2\beta^2 + 1} \quad \text{from (22)}$$

$$A < 1 \quad \text{from (51)}$$

After a minor rearrangement, we have a unified learning rate constraint as follows.

$$\eta \le \min\left\{ \frac{1}{2(\tau_{max} - 1)L}, \frac{1}{L\sqrt{2\tau_{max}(\tau_{max} - 1)(2\beta^2 + 1)}} \right\}$$

**Lemma 5.1.** *(Framework) Under Assumption* $1 \sim 3$*, if the learning rate satisfies* $\eta \le \frac{1}{2L}$*, FedLAMA ensures*

$$\frac{1}{K} \sum_{k=1}^{K} \mathbb{E}\left[ \|\nabla F(\mathbf{u}_k)\|^2 \right] \le \frac{2}{\eta K} \mathbb{E}\left[ F(\mathbf{u}_1) - F(\mathbf{u}_*) \right] + 2\eta L\sigma^2 \sum_{i=1}^{m} (p_i)^2 \tag{23}$$

$$+ \frac{L^2}{K} \sum_{k=1}^{K} \sum_{i=1}^{m} p_i \, \mathbb{E}\left[ \left\| \mathbf{u}_k - \mathbf{x}_k^i \right\|^2 \right].$$

*Proof.* Based on Assumption 1, we have

$$\mathbb{E}\left[ F(\mathbf{u}_{k+1}) - F(\mathbf{u}_k) \right] \le -\eta \underbrace{\mathbb{E}\left[ \langle \nabla F(\mathbf{u}_k), \sum_{i=1}^{m} p_i g_i(\mathbf{x}_k^i) \rangle \right]}_{T_1} + \frac{\eta^2 L}{2} \underbrace{\mathbb{E}\left[ \left\| \sum_{i=1}^{m} p_i g_i(\mathbf{x}_k^i) \right\|^2 \right]}_{T_2} \tag{24}$$

First, $T_1$ can be rewritten as follows.

$$T_1 = \mathbb{E}\left[ \langle \nabla F(\mathbf{u}_k), \sum_{i=1}^{m} p_i \left( g_i(\mathbf{x}_k^i) - \nabla F_i(\mathbf{x}_k^i) \right) \rangle \right] + \mathbb{E}\left[ \langle \nabla F(\mathbf{u}_k), \sum_{i=1}^{m} p_i \nabla F_i(\mathbf{x}_k^i) \rangle \right]$$

$$= \mathbb{E}\left[ \langle \nabla F(\mathbf{u}_k), \sum_{i=1}^{m} p_i \nabla F_i(\mathbf{x}_k^i) \rangle \right]$$

$$= \frac{1}{2} \|\nabla F(\mathbf{u}_k)\|^2 + \frac{1}{2} \mathbb{E}\left[ \left\| \sum_{i=1}^{m} p_i \nabla F_i(\mathbf{x}_k^i) \right\|^2 \right] - \frac{1}{2} \mathbb{E}\left[ \left\| \nabla F(\mathbf{u}_k) - \sum_{i=1}^{m} p_i \nabla F_i(\mathbf{x}_k^i) \right\|^2 \right], \tag{25}$$

where the last equality holds based on a basic equality: $2\mathbf{a}^\top \mathbf{b} = \|\mathbf{a}\|^2 + \|\mathbf{b}\|^2 - \|\mathbf{a} - \mathbf{b}\|^2$.

Then, $T_2$ can be bounded as follows.

$$T_2 = \mathbb{E}\left[ \left\| \sum_{i=1}^{m} p_i \left( g_i(\mathbf{x}_k^i) - \mathbb{E}\left[ g_i(\mathbf{x}_k^i) \right] \right) + \sum_{i=1}^{m} p_i \, \mathbb{E}\left[ g_i(\mathbf{x}_k^i) \right] \right\|^2 \right]$$

$$= \mathbb{E}\left[ \left\| \sum_{i=1}^{m} p_i \left( g_i(\mathbf{x}_k^i) - \nabla F_i(\mathbf{x}_k^i) \right) + \sum_{i=1}^{m} p_i \nabla F_i(\mathbf{x}_k^i) \right\|^2 \right]$$

$$\le 2 \mathbb{E}\left[ \left\| \sum_{i=1}^{m} p_i \left( g_i(\mathbf{x}_k^i) - \nabla F_i(\mathbf{x}_k^i) \right) \right\|^2 \right] + 2 \mathbb{E}\left[ \left\| \sum_{i=1}^{m} p_i \nabla F_i(\mathbf{x}_k^i) \right\|^2 \right]$$

$$= 2 \sum_{i=1}^{m} p_i^2 \, \mathbb{E}\left[ \left\| g_i(\mathbf{x}_k^i) - \nabla F_i(\mathbf{x}_k^i) \right\|^2 \right] + 2 \mathbb{E}\left[ \left\| \sum_{i=1}^{m} p_i \nabla F_i(\mathbf{x}_k^i) \right\|^2 \right]$$

$$\le 2\sigma^2 \sum_{i=1}^{m} p_i^2 + 2 \mathbb{E}\left[ \left\| \sum_{i=1}^{m} p_i \nabla F_i(\mathbf{x}_k^i) \right\|^2 \right], \tag{26}$$

where the last equality holds because $g_i(\mathbf{x}_k^i) - \nabla F_i(\mathbf{x}_k^i)$ has 0 mean and is independent across $i$, and the last inequality follows Assumption 3.

By plugging in (25) and (26) into (24), we have the following.

$$
\begin{aligned}
\mathbb{E}\left[F(\mathbf{u}_{k+1}) - F(\mathbf{u}_k)\right] \leq {}& -\frac{\eta}{2}\left\|\nabla F(\mathbf{u}_k)\right\|^2 - \frac{\eta}{2}\mathbb{E}\left[\left\|\sum_{i=1}^m p_i \nabla F_i(\mathbf{x}_k^i)\right\|^2\right] \\
& + \frac{\eta}{2}\mathbb{E}\left[\left\|\nabla F(\mathbf{u}_k) - \sum_{i=1}^m p_i \nabla F_i(\mathbf{x}_k^i)\right\|^2\right] + \eta^2 L\sigma^2 \sum_{i=1}^m p_i^2 \\
& + \eta^2 L\,\mathbb{E}\left[\left\|\sum_{i=1}^m p_i \nabla F_i(\mathbf{x}_k^i)\right\|^2\right] \\
= {}& -\frac{\eta}{2}\left\|\nabla F(\mathbf{u}_k)\right\|^2 - \frac{\eta}{2}(1 - 2\eta L)\,\mathbb{E}\left[\left\|\sum_{i=1}^m p_i \nabla F_i(\mathbf{x}_k^i)\right\|^2\right] \\
& + \frac{\eta}{2}\mathbb{E}\left[\left\|\nabla F(\mathbf{u}_k) - \sum_{i=1}^m p_i \nabla F_i(\mathbf{x}_k^i)\right\|^2\right] + \eta^2 L\sigma^2 \sum_{i=1}^m p_i^2
\end{aligned}
$$

If $\eta \leq \frac{1}{2L}$, it follows

$$
\begin{aligned}
\frac{\mathbb{E}\left[F(\mathbf{u}_{k+1}) - F(\mathbf{u}_k)\right]}{\eta} \leq {}& -\frac{1}{2}\left\|\nabla F(\mathbf{u}_k)\right\|^2 + \eta L\sigma^2 \sum_{i=1}^m p_i^2 \\
& + \frac{1}{2}\mathbb{E}\left[\left\|\nabla F(\mathbf{u}_k) - \sum_{i=1}^m p_i \nabla F_i(\mathbf{x}_k^i)\right\|^2\right] \\
\leq {}& -\frac{1}{2}\left\|\nabla F(\mathbf{u}_k)\right\|^2 + \eta L\sigma^2 \sum_{i=1}^m p_i^2 \qquad\qquad (27) \\
& + \frac{1}{2}\sum_{i=1}^m p_i\,\mathbb{E}\left[\left\|\nabla F_i(\mathbf{u}_k) - \nabla F_i(\mathbf{x}_k^i)\right\|^2\right] \\
\leq {}& -\frac{1}{2}\left\|\nabla F(\mathbf{u}_k)\right\|^2 + \eta L\sigma^2 \sum_{i=1}^m p_i^2 + \frac{L^2}{2}\sum_{i=1}^m p_i\,\mathbb{E}\left[\left\|\mathbf{u}_k - \mathbf{x}_k^i\right\|^2\right],
\end{aligned}
$$

where (27) holds based on the convexity of $\ell 2$ norm and Jensen's inequality.

By taking expectation and averaging across $K$ iterations, we have.

$$
\begin{aligned}
\frac{1}{K}\sum_{k=1}^K \frac{\mathbb{E}\left[F(\mathbf{u}_{k+1}) - F(\mathbf{u}_k)\right]}{\eta} \leq {}& -\frac{1}{2K}\sum_{k=1}^K \left\|\nabla F(\mathbf{u}_k)\right\|^2 + \eta L\sigma^2 \sum_{i=1}^m p_i^2 \\
& + \frac{L^2}{2K}\sum_{k-1}^K \sum_{i=1}^m p_i\,\mathbb{E}\left[\left\|\mathbf{u}_k - \mathbf{x}_k^i\right\|^2\right].
\end{aligned}
$$

After a minor rearrangement, we have a telescoping sum as follows.

$$\frac{1}{K}\sum_{k=1}^{K}\mathbb{E}\left[\|\nabla F(\mathbf{u}_k)\|^2\right] \leq \frac{2}{\eta K}\mathbb{E}\left[F(\mathbf{u}_1) - F(\mathbf{u}_{k+1})\right] + 2\eta L\sigma^2\sum_{i=1}^{m}p_i^2$$

$$+ \frac{L^2}{K}\sum_{k=1}^{K}\sum_{i=1}^{m}p_i\,\mathbb{E}\left[\|\mathbf{u}_k - \mathbf{x}_k^i\|^2\right]$$

$$\leq \frac{2}{\eta K}\mathbb{E}\left[F(\mathbf{u}_1) - F(\mathbf{u}_*)\right] + 2\eta L\sigma^2\sum_{i=1}^{m}p_i^2$$

$$+ \frac{L^2}{K}\sum_{k=1}^{K}\sum_{i=1}^{m}p_i\,\mathbb{E}\left[\|\mathbf{u}_k - \mathbf{x}_k^i\|^2\right],$$

where $\mathbf{u}_*$ indicates the local minimum. Here, we complete the proof. $\qquad\square$

**Lemma 5.2.** *(Model Discrepancy) Under Assumption $1 \sim 4$, if the learning rate satisfies $\eta < \frac{1}{2(\tau_{max}-1)L}$, FedLAMA ensures*

$$\frac{1}{K}\sum_{k=1}^{K}\sum_{i=1}^{m}p_i\,\mathbb{E}\left[\|\mathbf{u}_k - \mathbf{x}_k^i\|^2\right] \leq \frac{2\eta^2(\tau_{max}-1)\sigma^2}{1-A} + \frac{A\kappa^2}{L^2(1-A)}$$

$$+ \frac{A\beta^2}{KL^2(1-A)}\sum_{k=1}^{K}\mathbb{E}\left[\|\nabla F(\mathbf{u}_k)\|^2\right], \tag{28}$$

*where $A = 4\eta^2(\tau_{max}-1)^2L^2$ and $\tau_{max}$ is the largest averaging interval across all the layers.*

*Proof.* We begin with rewriting the weighted average of the squared distance using the vectorized form of the local models as follows.

$$\sum_{i=1}^{m}p_i\|\mathbf{u}_k - \mathbf{x}_k^i\|^2 = \sum_{i=1}^{m}\left\|\sqrt{p_i}\left(\mathbf{u}_k - \mathbf{x}_k^i\right)\right\|^2$$

$$= \|\mathbf{J}\hat{\mathbf{x}}_k - \hat{\mathbf{x}}_k\|^2 \tag{29}$$

$$= \|(\mathbf{J} - \mathbf{I})\hat{\mathbf{x}}_k\|^2,$$

where (29) holds by the commutative property of multiplication.

Then, according to the parameter update rule, we have

$$(\mathbf{J} - \mathbf{I})\hat{\mathbf{x}}_k = (\mathbf{J} - \mathbf{I})\mathbf{W}_{k-1}(\hat{\mathbf{x}}_{k-1} - \eta\hat{\mathbf{g}}_{k-1})$$

$$= (\mathbf{J} - \mathbf{I})\mathbf{W}_{k-1}\hat{\mathbf{x}}_{k-1} - (\mathbf{J} - \mathbf{W}_{k-1})\eta\hat{\mathbf{g}}_{k-1}, \tag{30}$$

where (30) holds because $\mathbf{JW} = \mathbf{J}$ based on the averaging matrix property 3, and $\mathbf{IW} = \mathbf{W}$.

Then, expanding the expression of $\mathbf{x}_{k-1}$, we have

$$(\mathbf{J} - \mathbf{I})\hat{\mathbf{x}}_k = (\mathbf{J} - \mathbf{I})\mathbf{W}_{k-1}(\mathbf{W}_{k-2}(\hat{\mathbf{x}}_{k-2} - \eta\hat{\mathbf{g}}_{k-2})) - (\mathbf{J} - \mathbf{W}_{k-1})\eta\hat{\mathbf{g}}_{k-1}$$

$$= (\mathbf{J} - \mathbf{I})\mathbf{W}_{k-1}\mathbf{W}_{k-2}\hat{\mathbf{x}}_{k-2} - (\mathbf{J} - \mathbf{W}_{k-1}\mathbf{W}_{k-2})\eta\hat{\mathbf{g}}_{k-2} - (\mathbf{J} - \mathbf{W}_{k-1})\eta\hat{\mathbf{g}}_{k-1}.$$

Repeating the same procedure for $\hat{\mathbf{x}}_{k-2}, \hat{\mathbf{x}}_{k-3}, \cdots, \hat{\mathbf{x}}_2$, we have

$$(\mathbf{J} - \mathbf{I})\hat{\mathbf{x}}_k = (\mathbf{J} - \mathbf{I})\prod_{s=1}^{k-1}\mathbf{W}_s\hat{\mathbf{x}}_1 - \eta\sum_{s=1}^{k-1}(\mathbf{J} - \prod_{l=s}^{k-1}\mathbf{W}_l)\hat{\mathbf{g}}_s$$

$$= -\eta\sum_{s=1}^{k-1}(\mathbf{J} - \prod_{l=s}^{k-1}\mathbf{W}_l)\hat{\mathbf{g}}_s, \tag{31}$$

where (31) holds because $\mathbf{x}_1^i$ is the same across all the workers and thus $(\mathbf{J} - \mathbf{I})\hat{\mathbf{x}}_1 = 0$.

Based on (31), we have

$$
\frac{1}{K} \sum_{k=1}^{K} \sum_{i=1}^{m} p_i \, \mathbb{E} \left[ \left\| \mathbf{u}_k - \mathbf{x}_k^i \right\|^2 \right]
$$

$$
= \frac{1}{K} \sum_{k=1}^{K} \left( \mathbb{E} \left[ \left\| (\mathbf{J} - \mathbf{I}) \hat{\mathbf{x}}_k \right\|^2 \right] \right)
$$

$$
= \frac{1}{K} \sum_{k=1}^{K} \left( \eta^2 \, \mathbb{E} \left[ \left\| \sum_{s=1}^{k-1} (\mathbf{J} - \prod_{l=s}^{k-1} \mathbf{W}_l) \hat{\mathbf{g}}_s \right\|^2 \right] \right)
$$

$$
= \frac{1}{K} \sum_{k=1}^{K} \left( \eta^2 \, \mathbb{E} \left[ \left\| \sum_{s=1}^{k-1} (\mathbf{J} - \prod_{l=s}^{k-1} \mathbf{W}_l)(\hat{\mathbf{g}}_s - \hat{\mathbf{f}}_s) + \sum_{s=1}^{k-1} (\mathbf{J} - \prod_{l=s}^{k-1} \mathbf{W}_l) \hat{\mathbf{f}}_s \right\|^2 \right] \right)
$$

$$
\leq \frac{2\eta^2}{K} \left( \underbrace{\sum_{k=1}^{K} \mathbb{E} \left[ \left\| \sum_{s=1}^{k-1} (\mathbf{J} - \prod_{l=s}^{k-1} \mathbf{W}_l)(\hat{\mathbf{g}}_s - \hat{\mathbf{f}}_s) \right\|^2 \right]}_{T_3} + \underbrace{\sum_{k=1}^{K} \mathbb{E} \left[ \left\| \sum_{s=1}^{k-1} (\mathbf{J} - \prod_{l=s}^{k-1} \mathbf{W}_l) \hat{\mathbf{f}}_s \right\|^2 \right]}_{T_4} \right)
$$

$$\tag{32}$$

where (32) holds based on the convexity of $\ell 2$ norm and Jensen's inequality. Now, we focus on bounding $T_3$ and $T_4$, separately.

**Bounding $T_3$**

$$
\sum_{k=1}^{K} \mathbb{E} \left[ \left\| \sum_{s=1}^{k-1} (\mathbf{J} - \prod_{l=s}^{k-1} \mathbf{W}_l)(\hat{\mathbf{g}}_s - \hat{\mathbf{f}}_s) \right\|^2 \right]
$$

$$
= \sum_{k=1}^{K} \sum_{s=1}^{k-1} \mathbb{E} \left[ \left\| (\mathbf{J} - \prod_{l=s}^{k-1} \mathbf{W}_l)(\hat{\mathbf{g}}_s - \hat{\mathbf{f}}_s) \right\|^2 \right] \tag{33}
$$

$$
\leq \sum_{k=1}^{K} \sum_{s=1}^{k-1} \mathbb{E} \left[ \left\| (\hat{\mathbf{g}}_s - \hat{\mathbf{f}}_s) \right\|^2 \left\| (\mathbf{J} - \prod_{l=s}^{k-1} \mathbf{W}_l) \right\|_{op}^2 \right], \tag{34}
$$

where (33) holds because $\hat{\mathbf{g}}_s - \hat{\mathbf{f}}_s$ has 0 mean and independent across $s$, and (34) holds based on Lemma A.1.

Without loss of generality, we replace $k$ with $a\tau_{max} + b$, where $a$ is the communication round index and $b$ is the iteration index within each communication round. Then, we have

$$\sum_{a=0}^{K/\tau_{max}-1} \sum_{b=1}^{\tau_{max}} \sum_{s=1}^{a\tau_{max}+b-1} \mathbb{E}\left[\left\|(\hat{\mathbf{g}}_s - \hat{\mathbf{f}}_s)\right\|^2 \left\|(\mathbf{J} - \prod_{l=s}^{k-1} \mathbf{W}_l)\right\|_{op}^2\right]$$

$$= \sum_{a=0}^{K/\tau_{max}-1} \sum_{b=1}^{\tau_{max}} \sum_{s=1}^{a\tau} \mathbb{E}\left[\left\|(\hat{\mathbf{g}}_s - \hat{\mathbf{f}}_s)\right\|^2 \left\|(\mathbf{J} - \prod_{l=s}^{a\tau_{max}+b-1} \mathbf{W}_l)\right\|_{op}^2\right]$$

$$+ \sum_{a=0}^{K/\tau_{max}-1} \sum_{b=1}^{\tau_{max}} \sum_{s=a\tau_{max}+1}^{a\tau_{max}+b-1} \mathbb{E}\left[\left\|(\hat{\mathbf{g}}_s - \hat{\mathbf{f}}_s)\right\|^2 \left\|(\mathbf{J} - \prod_{l=s}^{a\tau_{max}+b-1} \mathbf{W}_l)\right\|_{op}^2\right]$$

$$= \sum_{a=0}^{K/\tau_{max}-1} \sum_{b=1}^{\tau_{max}} \sum_{s=a\tau_{max}+1}^{a\tau_{max}+b-1} \mathbb{E}\left[\left\|(\hat{\mathbf{g}}_s - \hat{\mathbf{f}}_s)\right\|^2 \left\|(\mathbf{J} - \prod_{l=s}^{a\tau_{max}+b-1} \mathbf{W}_l)\right\|_{op}^2\right] \tag{35}$$

$$= \sum_{a=0}^{K/\tau_{max}-1} \sum_{b=1}^{\tau_{max}} \sum_{s=a\tau_{max}+1}^{a\tau+b-1} \mathbb{E}\left[\left\|(\hat{\mathbf{g}}_s - \hat{\mathbf{f}}_s)\right\|^2\right] \tag{36}$$

$$= \sum_{a=0}^{K/\tau_{max}-1} \sum_{b=1}^{\tau_{max}} \sum_{s=a\tau_{max}+1}^{a\tau_{max}+b-1} \sum_{i=1}^{m} p_i \mathbb{E}\left[\left\|(g_i(\mathbf{x}_s^i) - \nabla F_i(\mathbf{x}_s^i))\right\|^2\right]$$

$$\leq \sum_{a=0}^{K/\tau_{max}-1} \sum_{b=1}^{\tau_{max}} \sum_{s=a\tau_{max}+1}^{a\tau_{max}+b-1} \sum_{i=1}^{m} p_i \sigma^2 \tag{37}$$

$$= \sum_{a=0}^{K/\tau_{max}-1} \sum_{b=1}^{\tau_{max}} (b-1)\sigma^2 = \sum_{a=0}^{K/\tau_{max}-1} \frac{\tau_{max}(\tau_{max}-1)}{2}\sigma^2$$

$$\leq K\frac{(\tau_{max}-1)}{2}\sigma^2. \tag{38}$$

Remember FedLAMA synchronizes the whole parameters at least once after every $\tau_{max}$ iterations. Thus, (35) holds because $\prod_{l=s}^{a\tau_{max}+b-1} \mathbf{W}_l$ is $\mathbf{J}$ when $s \leq a\tau_{max}$, and thus $\mathbf{J} - \prod_{l=s}^{a\tau_{max}+b-1} \mathbf{W}_l$ becomes 0. (36) holds based on Lemma A.2. (37) holds based on Assumption 3.

**Bounding** $T_4$

$$\sum_{k=1}^{K-\tau_{max}} \mathbb{E}\left[\left\|\sum_{s=1}^{k-1}(\mathbf{J} - \prod_{l=s}^{k-1}\mathbf{W}_l)\hat{\mathbf{f}}_s\right\|^2\right]$$

$$= \sum_{a=0}^{K/\tau_{max}-1}\sum_{b=1}^{\tau_{max}} \mathbb{E}\left[\left\|\sum_{s=1}^{a\tau+b-1}(\mathbf{J} - \prod_{l=s}^{a\tau_{max}+b-1}\mathbf{W}_l)\hat{\mathbf{f}}_s\right\|^2\right]$$

$$= \sum_{a=0}^{K/\tau_{max}-1}\sum_{b=1}^{\tau_{max}} \mathbb{E}\left[\left\|\sum_{s=a\tau_{max}+1}^{a\tau_{max}+b-1}(\mathbf{J} - \prod_{l=s}^{a\tau_{max}+b-1}\mathbf{P}_l)\hat{\mathbf{f}}_s\right\|^2\right] \tag{39}$$

$$\leq \sum_{a=0}^{K/\tau_{max}-1}\sum_{b=1}^{\tau_{max}} \left((b-1)\sum_{s=a\tau_{max}+1}^{a\tau_{max}+b-1} \mathbb{E}\left[\left\|(\mathbf{J} - \prod_{l=s}^{a\tau_{max}+b-1}\mathbf{P}_l)\hat{\mathbf{f}}_s\right\|^2\right]\right) \tag{40}$$

$$\leq \sum_{a=0}^{K/\tau_{max}-1}\sum_{b=1}^{\tau_{max}} \left((b-1)\sum_{s=a\tau_{max}+1}^{a\tau_{max}+b-1} \mathbb{E}\left[\left\|\hat{\mathbf{f}}_s\right\|^2\left\|(\mathbf{J} - \prod_{l=s}^{a\tau_{max}+b-1}\mathbf{P}_l)\right\|_{op}^2\right]\right) \tag{41}$$

$$\leq \sum_{a=0}^{K/\tau_{max}-1}\sum_{b=1}^{\tau_{max}} \left((b-1)\sum_{s=a\tau_{max}+1}^{a\tau_{max}+b-1} \mathbb{E}\left[\left\|\hat{\mathbf{f}}_s\right\|^2\right]\right) \tag{42}$$

$$\leq \frac{\tau_{max}(\tau_{max}-1)}{2} \sum_{a=0}^{K/\tau_{max}-1} \left(\sum_{s=a\tau_{max}+1}^{a\tau_{max}+\tau_{max}-1} \mathbb{E}\left[\left\|\hat{\mathbf{f}}_s\right\|^2\right]\right)$$

$$\leq \frac{\tau_{max}(\tau_{max}-1)}{2} \sum_{k=1}^{K} \mathbb{E}\left[\left\|\hat{\mathbf{f}}_k\right\|^2\right]$$

$$= \frac{\tau_{max}(\tau_{max}-1)}{2} \sum_{k=1}^{K}\sum_{i=1}^{m} p_i\, \mathbb{E}\left[\left\|\nabla F_i(\mathbf{x}_k^i)\right\|^2\right], \tag{43}$$

where (39) holds because $\mathbf{J} - \prod_{l=s}^{a\tau_{max}+b-1}\mathbf{P}_l$ becomes 0 when $s \leq a\tau$. (40) holds based on the convexity of $\ell 2$ norm and Jensen's inequality. (41) holds based on Lemma A.1. (42) holds based on Lemma A.2.

### Final Result

By plugging in (38) and (43) into (32), we have

$$\frac{1}{K}\sum_{k=1}^{K}\sum_{i=1}^{m} p_i\, \mathbb{E}\left[\left\|\mathbf{u}_k - \mathbf{x}_k^i\right\|^2\right]$$

$$\leq \frac{2\eta^2}{K}\left(K\frac{(\tau_{max}-1)}{2}\sigma^2 + \frac{\tau_{max}(\tau_{max}-1)}{2}\left(\sum_{k=1}^{K}\sum_{i=1}^{m} p_i\, \mathbb{E}\left[\left\|\nabla F_i(\mathbf{x}_k^i)\right\|^2\right]\right)\right)$$

$$= \eta^2(\tau_{max}-1)\sigma^2 + \frac{\eta^2\tau_{max}(\tau_{max}-1)}{K}\left(\sum_{k=1}^{K}\sum_{i=1}^{m} p_i\, \mathbb{E}\left[\left\|\nabla F_i(\mathbf{x}_k^i)\right\|^2\right]\right) \tag{44}$$

The local gradient term on the right-hand side in (44) can be rewritten using the following inequality.

$$\mathbb{E}\left[\left\|\nabla F_i(\mathbf{x}_k^i)\right\|^2\right] = \mathbb{E}\left[\left\|\nabla F_i(\mathbf{x}_k^i) - \nabla F_i(\mathbf{u}_k) + \nabla F_i(\mathbf{u}_k)\right\|^2\right]$$

$$\leq 2\,\mathbb{E}\left[\left\|\nabla F_i(\mathbf{x}_k^i) - \nabla F_i(\mathbf{u}_k)\right\|^2\right] + 2\,\mathbb{E}\left[\left\|\nabla F_i(\mathbf{u}_k)\right\|^2\right] \tag{45}$$

$$\leq 2L^2\,\mathbb{E}\left[\left\|\mathbf{u}_k - \mathbf{x}_k^i\right\|^2\right] + 2\,\mathbb{E}\left[\left\|\nabla F_i(\mathbf{u}_k)\right\|^2\right], \tag{46}$$

where (45) holds based on the convexity of $\ell 2$ norm and Jensen's inequality.

Plugging in (46) into (44), we have

$$\frac{1}{K} \sum_{k=1}^{K} \sum_{i=1}^{m} p_i \, \mathbb{E}\left[\left\|\mathbf{u}_k - \mathbf{x}_k^i\right\|^2\right]$$

$$\leq \eta^2(\tau_{max} - 1)\sigma^2 + \frac{2\eta^2\tau_{max}(\tau_{max} - 1)L^2}{K} \sum_{k=1}^{K} \sum_{i=1}^{m} p_i \, \mathbb{E}\left[\left\|\mathbf{u}_k - \mathbf{x}_k^i\right\|^2\right]$$

$$+ \frac{2\eta^2\tau_{max}(\tau_{max} - 1)}{K} \sum_{k=1}^{K} \sum_{i=1}^{m} p_i \, \mathbb{E}\left[\left\|\nabla F_i(\mathbf{u}_k)\right\|^2\right] \tag{47}$$

After a minor rearranging, we have

$$\frac{1}{K} \sum_{k=1}^{K} \sum_{i=1}^{m} p_i \, \mathbb{E}\left[\left\|\mathbf{u}_k - \mathbf{x}_k^i\right\|^2\right] \leq \frac{\eta^2(\tau_{max} - 1)\sigma^2}{1 - 2\eta^2\tau_{max}(\tau_{max} - 1)L^2}$$

$$+ \frac{2\eta^2\tau_{max}(\tau_{max} - 1)}{K(1 - 2\eta^2\tau_{max}(\tau_{max} - 1)L^2)} \sum_{k=1}^{K} \sum_{i=1}^{m} p_i \, \mathbb{E}\left[\left\|\nabla F_i(\mathbf{u}_k)\right\|^2\right] \tag{48}$$

Let us define $A = 2\eta^2\tau_{max}(\tau_{max} - 1)L^2$. Then (48) is simplified as follows.

$$\frac{1}{K} \sum_{k=1}^{K} \sum_{i=1}^{m} p_i \, \mathbb{E}\left[\left\|\mathbf{u}_k - \mathbf{x}_k^i\right\|^2\right]$$

$$\leq \frac{\eta^2(\tau_{max} - 1)\sigma^2}{1 - A} + \frac{A}{KL^2(1 - A)} \sum_{k=1}^{K} \sum_{i=1}^{m} p_i \, \mathbb{E}\left[\left\|\nabla F_i(\mathbf{u}_k)\right\|^2\right]$$

Based on Assumption 4, we have

$$\frac{1}{K} \sum_{k=1}^{K} \sum_{i=1}^{m} p_i \, \mathbb{E}\left[\left\|\mathbf{u}_k - \mathbf{x}_k^i\right\|^2\right]$$

$$\leq \frac{\eta^2(\tau_{max} - 1)\sigma^2}{1 - A} + \frac{A\beta^2}{KL^2(1 - A)} \sum_{k=1}^{K} \mathbb{E}\left[\left\|\sum_{i=1}^{m} p_i \nabla F_i(\mathbf{u}_k)\right\|^2\right] + \frac{A\kappa^2}{L^2(1 - A)} \tag{49}$$

$$= \frac{\eta^2(\tau_{max} - 1)\sigma^2}{1 - A} + \frac{A\beta^2}{KL^2(1 - A)} \sum_{k=1}^{K} \mathbb{E}\left[\left\|\nabla F(\mathbf{u}_k)\right\|^2\right] + \frac{A\kappa^2}{L^2(1 - A)}, \tag{50}$$

where (50) holds based on the definition of the objective function (10).

Note that (49) is true only when $1 - A > 0$. Thus, after a minor rearrangement, we have a learning rate constraint as follows.

$$\eta < \frac{1}{2(\tau_{max} - 1)L} \tag{51}$$

Here, we complete the proof. $\qquad\square$

### A.1.3 PROOF OF OTHER LEMMAS

**Lemma A.1.** *Consider a real matrix $\mathbf{A} \in \mathbb{R}^{md_j \times md_j}$ and a real vector $\mathbf{b} \in \mathbb{R}^{md_j}$. If $\mathbf{b} \neq \mathbf{0}_{md_j}$, we have*

$$\|\mathbf{A}\mathbf{b}\| \leq \|\mathbf{A}\|_{op}\|\mathbf{b}\| \tag{52}$$

*Proof.*

$$\|\mathbf{Ab}\|^2 = \frac{\|\mathbf{Ab}\|^2}{\|\mathbf{b}\|^2}\|\mathbf{b}\|^2$$
$$\leq \|\mathbf{A}\|_{op}^2\|\mathbf{b}\|^2 \tag{53}$$

where (53) holds based on the definition of operator norm. □

**Lemma A.2.** *Suppose an $md \times md$ averaging matrix $\mathbf{P}$ and the full-averaging matrix $\mathbf{J}$, then*

$$\|\mathbf{J} - \mathbf{P}\|_{op}^2 = 1. \tag{54}$$

*regardless of which layers are chosen as the LCL.*

*Proof.* First, by the definition of averaging matrix $\mathbf{P}$, all the columns that do not correspond to the LCL are zeroed out in $\mathbf{J} - \mathbf{P}$. Then, based on the averaging matrix property 1 and 2, the remaining columns in $\mathbf{P}$ has 1 at all different rows. By the definition of $\mathbf{J}$, all the non-zero elements in $i^{th}$ column are the same $p_i, i \in \{1, \cdots, m\}$. Consequently, the remaining columns in $\mathbf{J} - \mathbf{P}$ are always orthogonal regardless of which layers are chosen as the LCL, and thus the eigenvalues of $\mathbf{J} - \mathbf{P}$ are either 1 or $-1$. Finally, by the definition of the matrix operator norm, $\|\mathbf{J} - \mathbf{P}\|_{op}^2 = max\{|\lambda(\mathbf{J} - \mathbf{P})|\} = 1$, where $\lambda(\cdot)$ indicates the eigenvalues of the input matrix. □

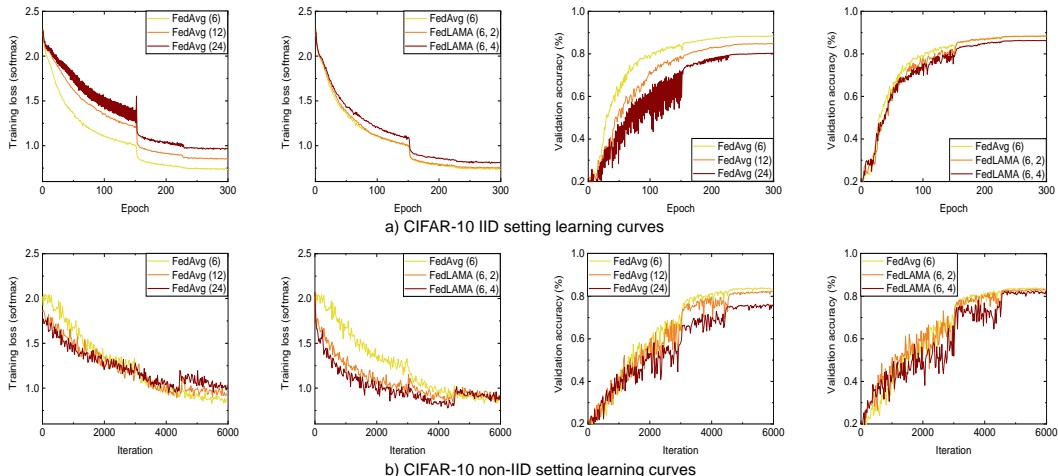

Figure 4: The learning curves of CIFAR-10 (ResNet20) training (128 clients). a): The curves for IID data distribution. b): The curves for non-IID data distribution ($\alpha = 0.1$). FedAvg ($x$) indicates FedAvg with the interval of $x$. FedLAMA ($x$, $y$) indicates FedLAMA with the base interval of $x$ and the interval increase factor of $y$. As the aggregation interval increases, FedAvg rapidly loses the convergence speed, and it results in achieving a lower validation accuracy within the fixed iteration budget. In contrast, FedLAMA effectively increases the aggregation interval while maintaining the convergence speed.

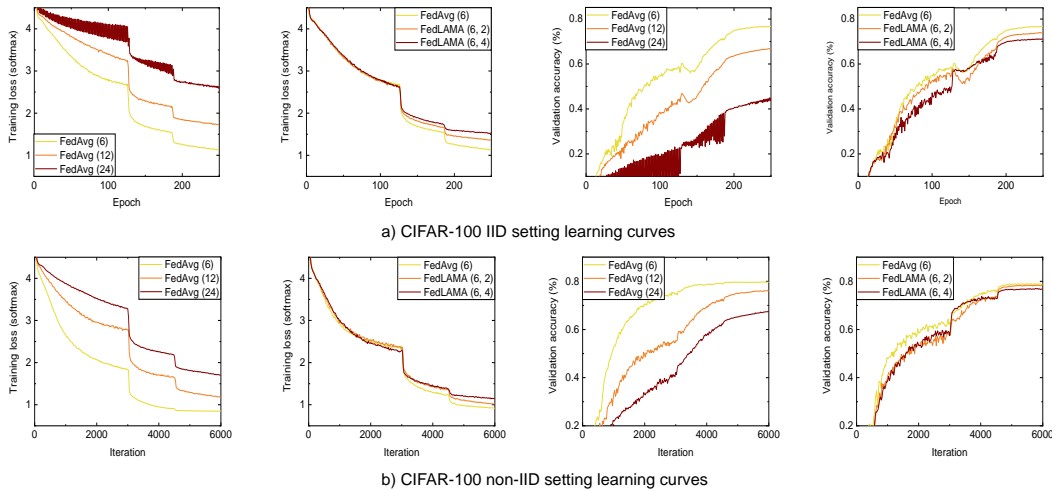

Figure 5: The learning curves of CIFAR-100 (WideResNet28-10) training (128 clients). a): The curves for IID data distribution. b): The curves for non-IID data distribution ($\alpha = 0.1$). FedAvg ($x$) indicates FedAvg with the interval of $x$. FedLAMA ($x$, $y$) indicates FedLAMA with the base interval of $x$ and the interval increase factor of $y$. While FedAvg significantly loses the convergence speed as the aggregation interval increases, FedLAMA has a marginl impact on it which results in a higher validation accuracy.

## A.2 ADDITIONAL EXPERIMENTAL RESULTS

In this section, we provide extra experimental results with extensive hyper-parameter settings. We commonly use 128 clients and a local batch size of 32 in all the experiments. The gradual learning rate warmup (Goyal et al. (2017)) is also applied to the first 10 epochs in all the experiments. Overall, the learning curve charts and the validation accuracy tables deliver the key insight that FedLAMA achieves a comparable convergence speed to the periodic full aggregation with the base interval

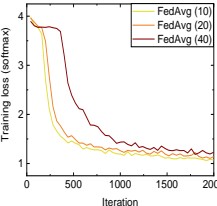 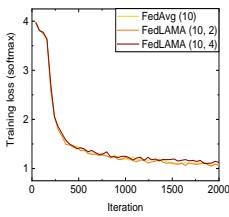 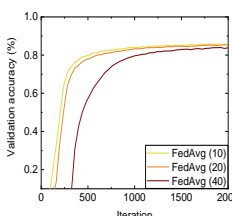 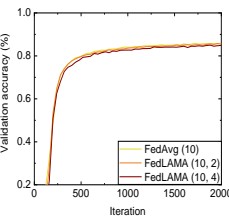

Figure 6: The learning curves of FEMNIST (CNN) training. FedAvg ($x$) indicates FedAvg with the interval of $x$. FedLAMA ($x, y$) indicates FedLAMA with the base interval of $x$ and the interval increase factor of $y$. FedLAMA curves are not strongly affected by the increased aggregation interval while FedAvg significantly loses the convergence speed as well as the validation accuracy.

($\tau$') while having the communication cost that is similar to the periodic full aggregation with the increased interval ($\phi\tau'$).

**Artificial Data Heterogeneity** – For CIFAR-10 and CIFAR-100, we artificially generate the heterogeneous data distribution using Dirichlet's distribution. The concentration coefficient $\alpha$ is set to 0.1, 0.5, and 1.0 to evaluate the performance of FedLAMA across a variety of degree of data heterogeneity. Note that the small concentration coefficient represents the highly heterogeneous numbers of local samples across clients as well as the balance of the samples across the labels. We used the data distribution source code provided by FedML (He et al. (2020)).

**CIFAR-10** – Figure 4 shows the full learning curves for IID and non-IID CIFAR-10 datasets. The hyper-parameter settings correspond to Table 4 and 1. First, as the aggregation interval increases from 6 to 24, FedAvg suffers from the slower convergence, and it results in achieving a lower validation accuracy, regardless of the data distribution. In contrast, FedLAMA learning curves are marginally affected by the increased aggregation interval. Table 6 and 7 show the CIFAR-10 classification performance of FedLAMA across different $\phi$ settings. As expected, the accuracy is reduced as $\phi$ increases. The IID and non-IID data settings show the common trend. Depending on the system network bandwidth, $\phi$ can be tuned to be an appropriate value. When $\phi = 2$, the accuracy is almost the same as or even slightly higher than FedAvg accuracy. If the network bandwidth is limited, one can increase $\phi$ and slightly increase the epoch budget to achieve a good accuracy. Table 8 shows the CIFAR-10 accuracy across different $\tau'$ settings. We see that the accuracy is significantly dropped as $\tau'$ increases.

**CIFAR-100** – Figure 5 shows the learning curves for IID and non-IID CIFAR-100 datasets. Likely to CIFAR-10 results, FedAvg learning curves are strongly affected as the aggregation interval increases from 6 to 24 while FedLAMA learning curves are not strongly affected. Table 9 and 10 show the CIFAR-100 classification performance of FedLAMA across different $\phi$ settings. FedLAMA achieves a comparable accuracy to FedAvg with a short aggregation interval, even when the degree of data heterogeneity is extreamly high (25% device sampling and Direchlet's coefficient of 0.1). Table 11 shows the FedAvg accuracy with different $\tau'$ settings. Under the strongly heterogeneous data distributions, FedAvg with a large aggregation interval ($\tau \geq 12$) do not achieve a reasonable accuracy.

**FEMNIST** – Figure 6 shows the learning curves of CNN training. Likely to the previous two datasets, the periodic full aggregation suffers from the slower convergence as the aggregation interval increases. FedLAMA learning curves are not much affected by the increased aggregation interval, and it results in achieving a higher validation accuracy after the same number of iterations. Table 12 shows the FEMNIST classification performance of FedLAMA across different $\phi$ settings. FedLAMA achieves a similar accuracy to the baseline (FedAvg with $\tau' = 10$) even when using a large interval increase factor $\phi \geq 4$. These results demonstrate the effectiveness of the proposed layer-wise adaptive model aggregation method on the problems with heterogeneous data distributions.

Table 6: (IID data) CIFAR-10 classification results of FedLAMA with different $\phi$ settings.

| # of clients | Local batch size | LR | Averaging interval: $\tau'$ | Interval increase factor: $\phi$ | Validation acc. |
|---|---|---|---|---|---|
| 128 | 32 | 0.8 | 6 | 1 (FedAvg) | $88.37 \pm 0.1\%$ |
| | | | | 2 | $88.41 \pm 0.04\%$ |
| | | 0.5 | | 4 | $86.33 \pm 0.2\%$ |
| | | | | 8 | $85.08 \pm 0.04\%$ |

Table 7: (Non-IID data) CIFAR-10 classification results of FedLAMA with different $\phi$ settings.

| # of clients | Local batch size | LR | $\tau'$ | Active ratio | Dirichlet coeff. | $\phi$ | Validation acc. |
|---|---|---|---|---|---|---|---|
| 128 | 32 | 0.8 | 6 | 100% | 1 | 1 (FedAvg) | $90.79 \pm 0.1\%$ |
| | | | | | | 2 | $89.01 \pm 0.04\%$ |
| | | | | | | 4 | $87.84 \pm 0.01\%$ |
| | | | | 100% | 0.5 | 1 (FedAvg) | $90.53 \pm 0.18\%$ |
| | | | | | | 2 | $89.21 \pm 0.2\%$ |
| | | | | | | 4 | $86.68 \pm 0.12\%$ |
| | | | | 100% | 0.1 | 1 (FedAvg) | $89.52 \pm 0.11\%$ |
| | | | | | | 2 | $89.00 \pm 0.1\%$ |
| | | | | | | 4 | $84.82 \pm 0.08\%$ |
| | | | | 50% | 1 | 1 (FedAvg) | $90.34 \pm 0.12\%$ |
| | | | | | | 2 | $89.56 \pm 0.13\%$ |
| | | | | | | 4 | $87.48 \pm 0.21\%$ |
| | | | | 50% | 0.5 | 1 (FedAvg) | $89.86 \pm 0.13\%$ |
| | | | | | | 2 | $88.44 \pm 0.15\%$ |
| | | | | | | 4 | $87.29 \pm 0.18\%$ |
| | | | | 50% | 0.1 | 1 (FedAvg) | $87.83 \pm 0.2\%$ |
| | | | | | | 2 | $87.40 \pm 0.17\%$ |
| | | | | | | 4 | $85.92 \pm 0.21\%$ |
| | | 0.6 | | 25% | 1 | 1 (FedAvg) | $88.97 \pm 0.03\%$ |
| | | | | | | 2 | $87.89 \pm 0.2\%$ |
| | | | | | | 4 | $86.61 \pm 0.1\%$ |
| | | | | 25% | 0.5 | 1 (FedAvg) | $87.59 \pm 0.05\%$ |
| | | | | | | 2 | $87.12 \pm 0.08\%$ |
| | | | | | | 4 | $86.57 \pm 0.02\%$ |
| | | 0.3 | | 25% | 0.1 | 1 (FedAvg) | $84.02 \pm 0.04\%$ |
| | | | | | | 2 | $83.55 \pm 0.02\%$ |
| | | | | | | 4 | $83.06 \pm 0.03\%$ |

Table 8: (Non-IID data) CIFAR-10 classification results of FedAvg with different $\tau'$ settings.

| # of clients | Local batch size | LR | $\tau'$ | Active ratio | Dirichlet coeff. | $\phi$ | Validation acc. |
|---|---|---|---|---|---|---|---|
| 128 | 32 | 0.8 | 6 | 100% | 0.1 | 1 (FedAvg) | $89.52 \pm 0.11\%$ |
| | | | 12 | | | 1 (FedAvg) | $87.29 \pm 0.05\%$ |
| | | | 24 | | | 1 (FedAvg) | $84.82 \pm 0.1\%$ |
| 128 | 32 | 0.3 | 6 | 25% | 0.1 | 1 (FedAvg) | $84.02 \pm 0.1\%$ |
| | | | 12 | | | 1 (FedAvg) | $82.48 \pm 0.2\%$ |
| | | | 24 | | | 1 (FedAvg) | $76.72 \pm 0.1\%$ |

Table 9: (IID data) CIFAR-100 classification results of FedLAMA with different $\phi$ settings.

| # of clients | Local batch size | LR | Averaging interval: $\tau'$ | Interval increase factor: $\phi$ | Validation acc. |
|---|---|---|---|---|---|
| 128 | 32 | 0.6 | 6 | 1 (FedAvg) | $76.50 \pm 0.02\%$ |
| | | | | 2 | $75.99 \pm 0.03\%$ |
| | | | | 4 | $76.17 \pm 0.2\%$ |
| | | | | 8 | $76.15 \pm 0.2\%$ |

Table 10: (Non-IID data) CIFAR-100 classification results of FedLAMA with different $\phi$ settings.

| # of clients | Local batch size | LR | $\tau'$ | Active ratio | Dirichlet coeff. | $\phi$ | Validation acc. |
|---|---|---|---|---|---|---|---|
| 128 | 32 | 0.4 | 6 | 100% | 1 | 1 (FedAvg) | $80.34 \pm 0.01\%$ |
| | | | | | | 2 | $78.92 \pm 0.01\%$ |
| | | | | | | 4 | $77.16 \pm 0.05\%$ |
| | | | | 100% | 0.5 | 1 (FedAvg) | $80.19 \pm 0.02\%$ |
| | | | | | | 2 | $78.88 \pm 0.1\%$ |
| | | | | | | 4 | $78.03 \pm 0.08\%$ |
| | | 0.2 | | 100% | 0.1 | 1 (FedAvg) | $79.78 \pm 0.02\%$ |
| | | | | | | 2 | $79.07 \pm 0.02\%$ |
| | | | | | | 4 | $79.32 \pm 0.01\%$ |
| | | 0.4 | | 50% | 1 | 1 (FedAvg) | $79.94 \pm 0.1\%$ |
| | | | | | | 2 | $78.98 \pm 0.01\%$ |
| | | | | | | 4 | $77.50 \pm 0.02\%$ |
| | | | | 50% | 0.5 | 1 (FedAvg) | $79.95 \pm 0.05\%$ |
| | | | | | | 2 | $78.37 \pm 0.05\%$ |
| | | | | | | 4 | $76.93 \pm 0.1\%$ |
| | | 0.2 | | 50% | 0.1 | 1 (FedAvg) | $79.62 \pm 0.06\%$ |
| | | | | | | 2 | $78.76 \pm 0.02\%$ |
| | | | | | | 4 | $77.44 \pm 0.02\%$ |
| | | 0.4 | | 25% | 1 | 1 (FedAvg) | $78.78 \pm 0.02\%$ |
| | | | | | | 2 | $78.10 \pm 0.02\%$ |
| | | 0.2 | | | | 4 | $76.84 \pm 0.03\%$ |
| | | | | 25% | 0.5 | 1 (FedAvg) | $78.81 \pm 0.01\%$ |
| | | | | | | 2 | $77.86 \pm 0.04\%$ |
| | | 0.4 | | | | 4 | $77.01 \pm 0.1\%$ |
| | | | | 25% | 0.1 | 1 (FedAvg) | $79.06 \pm 0.03\%$ |
| | | | | | | 2 | $78.63 \pm 0.02\%$ |
| | | 0.2 | | | | 4 | $77.17 \pm 0.01\%$ |

Table 11: (Non-IID data) CIFAR-100 classification results of FedAvg with different $\tau'$ settings.

| # of clients | Local batch size | LR | $\tau'$ | Active ratio | Dirichlet coeff. | $\phi$ | Validation acc. |
|---|---|---|---|---|---|---|---|
| 128 | 32 | 0.4 | 6 | 100% | 0.1 | 1 (FedAvg) | $79.78 \pm 0.02\%$ |
| | | | 12 | | | 1 (FedAvg) | $77.71 \pm 0.1\%$ |
| | | | 24 | | | 1 (FedAvg) | $69.63 \pm 0.1\%$ |
| 128 | 32 | 0.4 | 6 | 25% | 0.1 | 1 (FedAvg) | $79.06 \pm 0.03\%$ |
| | | | 12 | | | 1 (FedAvg) | $76.16 \pm 0.05\%$ |
| | | | 24 | | | 1 (FedAvg) | $67.43 \pm 0.1\%$ |

Table 12: FEMNIST classification results of FedLAMA with different $\phi$ settings.

| # of clients | Local batch size | LR | Averaging interval: $\tau'$ | Active ratio | Interval increase factor: $\phi$ | Validation acc. |
|---|---|---|---|---|---|---|
| 128 | 32 | 0.04 | 12 | 100% | 1 (FedAvg) | $85.74 \pm 0.21\%$ |
| | | | | | 2 | $85.40 \pm 0.13\%$ |
| | | | | | 4 | $84.67 \pm 0.1\%$ |
| | | | | | 8 | $84.15 \pm 0.18\%$ |
| | | | | 50% | 1 (FedAvg) | $86.59 \pm 0.2\%$ |
| | | | | | 2 | $86.07 \pm 0.1\%$ |
| | | | | | 4 | $85.77 \pm 0.15\%$ |
| | | | | | 8 | $85.31 \pm 0.03\%$ |
| | | | | 25% | 1 (FedAvg) | $86.04 \pm 0.2\%$ |
| | | | | | 2 | $86.01 \pm 0.1\%$ |
| | | | | | 4 | $85.62 \pm 0.08\%$ |
| | | | | | 8 | $85.23 \pm 0.1\%$ |

