# OpenReview forum: "Layer-wise Adaptive Model Aggregation for Scalable Federated Learning"
_ICLR.cc/2022/Conference — ICLR 2022 Submitted_

### Official Review · Reviewer_graH · 2021-10-24

**Correctness:** 3
**Technical Novelty And Significance:** 3
**Empirical Novelty And Significance:** 3
**Recommendation:** 8
**Confidence:** 3

**Main Review:**

Effectively reducing the communication overhead is an important problem for federated learning, with the recent advance of layer-wise model freezing training, this paper manages to extend this idea for network traffic reduction in federated learning. Concretely, I list the main strengths and minor weakness of this paper as below:

Advantages:

+ To leverage layer-wise model freezing for federated learning is a simple but novel idea. According to the empirical study, this method seems to be effective and efficient.
+ The paper also includes tight convergence analysis for the proposed method.

Disadvantages:
+ The communication cost calculated in Section 6 is a little sloppy, in practices, there are plenty of system advances managing to hide the communication overhead within the computation slots, e.g. PyTorch-DDP (https://arxiv.org/abs/2006.15704). In this fashion, the actual runtime may not be simply linear to the amount of network traffic. Perhaps, a more sophisticated cost model would consider both the computation power of the edge device and network condition during training, then model the cost including the mechanism of overlapping communication and computation to estimate the actual runtime.



**Summary Of The Paper:**

This paper proposes a novel adaptive model aggregating schema under a federated learning setting, where it effective reduces the communication overhead without significantly sacrificing the generalization performance. Both empirical study and theoretical analysis are included to justify the effectiveness and efficiency of the proposed method.

**Summary Of The Review:**

The idea of utilizing layer-wise model freezing for federated learning is an interesting idea, the paper includes reasonable empirical study and convergence analysis to illustrate the effectiveness of this proposed approach.
Although some part of the cost model in the simulation is a little sloppy, I tend to advocate the acceptance of this paper.

---

> ### Author Response · Authors · 2021-11-12
> **Clarification of the question**
>
> We appreciate the comments. Please find our response below.
>
> 1. *Perhaps, a more sophisticated cost model would consider both the computation power of the edge device and network condition during training, then model the cost including the mechanism of overlapping communication and computation to estimate the actual runtime.*
>
> **[Problem scope]**: In this paper, we narrowed down the scope of our study to network bandwidth consumption and focused on how to reduce the consumption by adjusting the model aggregation frequency at each layer while maintaining the convergence property. Indeed, the reduced communication frequency implies that the degree of overlap between communication and computation can be improved (not always depending on which layers are chosen to be aggregated less frequently). As explained in the paper, the worst-case latency cost of FedLAMA is the same as the periodic full averaging with the base interval. Therefore, we think the current relatively simple communication cost modeling is sufficient for our study. We consider improving the system-wise performance of FedLAMA to be interesting future work.

---

### Official Review · Reviewer_9DWJ · 2021-10-30

**Correctness:** 3
**Technical Novelty And Significance:** 2
**Empirical Novelty And Significance:** 2
**Recommendation:** 5
**Confidence:** 5

**Main Review:**

They provide a communication-efficient federated optimization and relax the aggregation frequency by quantifying the model discrepancy between local models and global models and adaptively adjusting the aggregation interval in a layer-wise manner.

Some questions are the following.

1）The baseline is FedAvg, too few. Why not compare with other the periodic full aggregation scheme (e.g., FedProx, FedNova, SCAFFOLD, etc.), even some synchronized schemes? the synchronized schemes refer to,

[1] Chen, Yang, Xiaoyan Sun, and Yaochu Jin. "Communication-efficient federated deep learning with
layerwise asynchronous model update and temporally weighted aggregation." IEEE transactions on neural
networks and learning systems 31.10 (2019): 4229-4238.

[2] Xie, C., Koyejo, S., & Gupta, I. (2019). Asynchronous federated optimization. arXiv preprint
arXiv:1903.03934.

2）FedLAMA seems to lead to some accuracy drop as the experiments show. They do not show the total time in terms of communication cost, just show the reduced ratio for communication. It is not clear about absolute earnings with sacrificing the accuracy. In some cases, especially in industry, the model's accuracy matters more than training time (e.g., the total training time is within several hours.)

**Summary Of The Paper:**

In this paper, the authors propose a layer-wise model aggregation scheme in federated learning cases to reduce the communication cost. Specifically, they quantified the model discrepancy between local models and global models and adaptively adjusted the aggregation interval in a layer-wise manner. By increasing the aggregation intervals and relaxing the aggregation frequency, the method can reduce the communication cost in federated learning cases. The experimental results show it can reduce the communication cost for IID data and non-IID data compared to FedAvg.

**Summary Of The Review:**

Thanks to the authors for their efforts for their theoretical analysis and experimentally proof of the proposed federated optimization scheme. The experimental results are a little weak to show the effectiveness and the practicality.

---

> ### Author Response · Authors · 2021-11-12
> **Clarified the reviewer's concerns.**
>
> We appreciate the valuable comments. We wrote our responses to the comments as follows. Overall, we think the reviewer's concerns are mostly based on the misunderstanding of our research scope and goal. We have clarified each concern as below. The paper and the supplementary material have been accordingly updated.
>
> 1. *The baseline is FedAvg, too few. Why not compare with other the periodic full aggregation scheme (e.g., FedProx, FedNova, SCAFFOLD, etc.), even some synchronized schemes?*
>
> **[Comparison criteria]**: Our baseline is the periodic full model averaging rather than FedAvg. We believe that the valid evaluation should be the comparison across different ‘model aggregation’ schemes, not across different ‘optimizer’s. Thus, we used the same vanilla local SGD as a local solver for both the baseline and our proposed algorithm and then compared the performance across different model aggregation algorithms. As explained in the paper, the proposed FedLAMA algorithm can be applied to any of the advanced optimizers, such as FedProx or FedNova, without having any conflicts. This fact proves that the other optimizers are not the baselines to be compared with but the complementary applications. To make things clear, we elaborated on this discussion in the paper and uploaded it again. We do consider harmonizing FedLAMA with other optimizers as promising future work.
>
> 2. *FedLAMA seems to lead to some accuracy drop as the experiments show. They do not show the total time in terms of communication cost, just show the reduced ratio for communication. It is not clear about absolute earnings with sacrificing the accuracy.*
>
> **[The performance gain of FedLAMA]**: First, it cannot be simply considered as ‘dropping’ the accuracy. FedLAMA finds the layer-wise aggregation interval settings that minimize the loss in statistical efficiency while maximizing the scaling efficiency (the reduced number of communications). As the aggregation interval increases, the convergence bound monotonically increases. That said, we rather focus on how to increase the interval while having a marginal impact on the convergence rate.
>
> Since we compare the accuracy achieved within a fixed same iteration budget across different model aggregation schemes, the reduced number of communications directly means the scaling performance improvement. In addition, we 'simulate' federated learning due to the limited resources. Such a simulation on fewer GPUs than the actual clients is a common experimental setting in the Federated Learning research community. Under this setting, we believe the number of communications is a better metric for evaluation than the wall-clock time.

---

### Official Review · Reviewer_YGZf · 2021-11-03

**Correctness:** 1
**Technical Novelty And Significance:** 2
**Empirical Novelty And Significance:** 2
**Recommendation:** 5
**Confidence:** 5

**Main Review:**

Strengths:

1. It is an interesting idea to adjust the aggregation interval for layer-wise aggregation.
2. Detailed theoretical analysis and experiments are conducted.
3. The paper is well-written.


Weaknesses:

1. It is uncertain that the proposed method can really reduce communication costs. Due to the reduced frequency of aggregation and adopt layer-wise partial model aggregation strategy in each round, it is very likely that the proposed method will decrease the convergence speed of the global model. Therefore, the proposed method needs more communication rounds to achieve a convergence state, thus the total communication costs are still very high. Moreover, at each communication round, the client still needs to download the full model from the server. Therefore, the downloading communication costs haven't been improved.
2. In the convergence analysis, Assumption 4 is a little tricky, and helps to avoid much detailed and further analysis. The convergence rate is O(1/m^½ K^½), which is loose compared to some convergence analysis of FedAvg, such as ON THE CONVERGENCE OF FEDAVG ON NON-IID DATA (https://arxiv.org/abs/1907.02189).
3. The experiment is insufficient to support their claims. For example, the only baseline is FedAvg, and it is necessary to choose some compression methods as a baseline. Moreover, the communication cost should consider the number of communication rounds to achieve a convergence state.


Questions and suggestions:

1. Algorithm 1 should explicitly state which steps are executed on the server or client. For example, line 3 should be on the client, and lines from 4 to 10 are on the server. Moreover, it should explicitly highlight when the communication occurs.
2. It is necessary to draw the convergence curves of the proposed methods and baselines.
3. It is unclear how to construct non-IID scenarios in the experiment setting.
4. There are some federated learning methods that only update parts of its model. This paper should also consider these baseline methods.

http://proceedings.mlr.press/v139/collins21a/collins21a.pdf

https://arxiv.org/abs/2102.07623

**Summary Of The Paper:**

This paper proposes an adaptive interval schema for layer-wise model aggregation in federated settings. The aim of the proposed method is to reduce communication costs by adaptively decreasing the frequency of layer-wise aggregation with consideration of model discrepancy.

**Summary Of The Review:**

This paper proposes an interesting layer-wise model aggregation method to reduce communication cost while trying to keep the same performance with FedAvg. However, the proposed method’s design, theoretical analysis and experiments cannot support its claim.

---

> ### Author Response · Authors · 2021-11-12
> **Clarifications of the concerns.**
>
> Thank you for the thorough review of our paper. We hope our responses clarify the main goal of our work and resolve the reviewer’s concerns. The paper and supplementary material have been updated accordingly (we added the learning curves).
>
> 1. *It is uncertain that the proposed method can really reduce communication costs*.
>
> **[The best trade-off between scaling efficiency and statistical efficiency]**: We do not argue that FedLAMA simply ‘reduces’ the communication cost as compared to the general periodic full aggregation scheme. FedLAMA aims to find the layer-wise interval settings that minimize the communication frequency while not strongly affecting the convergence property. As compared to the periodic full averaging with the base interval $\tau'$, FedLAMA increases the aggregation interval at some layers, and thus the overall convergence bound is increased. However, FedLAMA finds the settings that minimize such a loss in statistical efficiency by choosing the low-priority layers and increasing their aggregation intervals. While having a marginally increased convergence bound, the communication cost is dramatically reduced thanks to the increased aggregation interval.
>
> Our experimental results show that the increased convergence bound does not significantly harm the accuracy in many different settings. In some cases, it achieves almost the same or even higher accuracy than the baseline while enjoying the reduced number of communications. Because we compare the accuracy achieved within the same fixed iteration budget, the reduced number of communications directly means the improved scaling efficiency.
>
>  2. *In the convergence analysis, Assumption 4 is a little tricky, and helps to avoid much detailed and further analysis*.
>
> **[More general assumption]**: This dissimilarity assumption is used in recent Federated Learning literature [1,2]. We do know that some other works assume the bounded variance across the clients. That assumption can be considered to be stronger than ours because they do not consider the bias across the clients. In that sense, this assumption is reasonable enough to analyze the performance of FedLAMA.
>
> 3. *The convergence rate is O(1/m^½ K^½), which is loose compared to some convergence analysis of FedAvg*.
>
> **[Our analysis considers non-convex problems]**: First, *Li et al., On the Convergence of FedAvg on Non-IID Data* provide the convergence analysis of strongly-convex problems. Because our analysis considers non-convex problems that include neural network training, their results cannot be directly compared to ours. In addition, the main research goal of this paper is not to improve the theoretical convergence property. We focus on exploring a novel layer-wise model aggregation scheme and showing that our algorithm provides a solid convergence guarantee and linear speedup like other federated optimizers. Considering the goal of our work, thus, we believe our analysis sufficiently supports the proposed algorithm.
>
> 4. *Non-IID is not explained*.
>
> **[Added in Appendix]**: We explained the details about how we generated the artificial data heterogeneity using Dirichlet distribution in Appendix.
>
> 5. *The experiment is insufficient to support their claims. For example, the only baseline is FedAvg, and it is necessary to choose some compression methods as a baseline*.
>
> **[The appropriate baselines]**: Our focus is on the model aggregation scheme in Federated Learning. Thus, we believe that the valid evaluation is the performance comparison across different model aggregation schemes. When comparing the performance in our experiments, thus, we keep all the other factors the same, such as optimizers, the number of clients, the local batch size, and only change the model aggregation schemes. We rather consider the compression techniques as good complementary applications. One can consider adopting FedLAMA and compressing the local update in a layer-wise manner within our proposed algorithm. Because these two techniques address the communication cost issue at different angles, they can be adopted together without any conflicts. We do consider harmonizing FedLAMA with other communication-efficient methods as promising future work.
>
> 6. *The communication round complexity should be presented*.
>
> **[Clarification]**: Because our proposed model aggregation scheme does not fully aggregate the whole parameters at every communication round, its communication round complexity cannot be directly compared to that of the conventional periodic full averaging scheme. Thus, we provide the iteration-wise complexity in our analysis.
>
> [1] Wang et al., Tackling the Objective Inconsistency Problem in Heterogeneous Federated Optimization
>
> [2] Karimireddy et al., SCAFFOLD: Stochastic Controlled Averaging for Federated Learning

---

> > ### Comment · Reviewer_YGZf · 2021-11-30
> > **Thanks for the response.**
> >
> > I am not convinced that the communication cost is a strong motivation to conduct layer-wise aggregation in federated learning. In particular, the experiment cannot show a significant improvement in communication efficiency. Therefore, finding the trade-off between scaling efficiency and statistical efficiency is impractical in real applications.
> >
> > Although this paper cannot provide an entire solution with strong practical motivation, I could buy the technique idea for the layer-wise aggregation in federated learning. This layer-wise solution in FL could potentially pave the way towards something that we cannot forecast at this moment. Therefore, I could increase my score to 5 to support the idea for layer-wise aggregation in FL.

---

### Official Review · Reviewer_7ks6 · 2021-11-03

**Correctness:** 3
**Technical Novelty And Significance:** 2
**Empirical Novelty And Significance:** 2
**Recommendation:** 5
**Confidence:** 4

**Main Review:**

Pros:

1. The problem studied is important and the idea is novel.
2. This paper provides the convergence rate.
3. This paper conducts extensive experiments.

Cons:
1. There are some errors in theoretical analysis. Eq.(15) is not correct. Eq.(25) is also wrong.

2. For Lemma 5.2, there should be a constraint for $\eta$.

3. The convergence upper bound is worse than standard FedAvg. In Remark 1, when $K=O(m^3)$, FedLAMA can achieve linear speedup. However, in [1], $K=O(m)$. Thus, this bound is much worse than [1]


[1] On the Linear Speedup Analysis of Communication Efficient Momentum SGD for Distributed Non-Convex Optimization

**Summary Of The Paper:**

This paper developed an adaptive aggregation method for federated learning. The theoretical analysis shows how the interval affects the convergence rate. The experiments show that it can reduce the communication cost.

**Summary Of The Review:**

This paper proposed a new idea for reducing communication costs. But the theoretical analysis has some flaws.

---

> ### Author Response · Authors · 2021-11-12
> **Corrected errors in the theoretical analysis.**
>
> Thank you for pointing out the errors in our analysis. We have addressed all the errors after thorough checks. Fortunately, the errors do not affect the final results of our analysis (they were caused by a wrong definition of the scaled average gradients). We have uploaded the corrected version of the supplementary material. Now we believe that our analysis does not have any flaws and would like the reviewers to consider the score again.
>
> 1. *There are some errors in theoretical analysis. Eq. (15) is not correct. Eq. (25) is also wrong*.
>
> **[Corrected errors]**: Thank you for pointing out the errors in the proof. We found that the definition of the weighted-vectorized gradients was wrong, and have corrected them. So, the equation (Eq. 27 in the updated version) is now true. We also revised all the relevant parts in the preliminary and other proofs.
>
> 2. *For Lemma 5.2, there should be a constraint for the learning rate*.
>
> **[No learning rate constraints in Lemma 5.2]**: We thoroughly reviewed our proofs and concluded that there are no learning rate constraints in Lemma 5.2. Theorem 1 has two learning rate constraints: one comes from Lemma 5.1 as we specified and the other comes after plugging in Lemma 5.2 into the framework when simplifying (21) on page 15. We summarized both constraints in Theorem 1 proof.
>
> 3. *The convergence upper bound is worse than standard FedAvg. In Remark 1, when $K=\mathcal{O}(m^3)$, FedLAMA can achieve linear speedup. However, in [1], $K=\mathcal{O}(m)$. Thus, this bound is much worse than [1]*.
>
> **[The complexity is the same]**: In the reference work the reviewer provided [1], their linear speedup condition is $K = \mathcal{O}(m^3)$, not $\mathcal{O}(m)$. Their corollary 2 (page 6) shows such a condition. When $\kappa$ is not 0 (non-iid), ignoring all the other constants such as $\beta$ and Lipschitz constant, the averaging interval $I$ becomes larger than 1 only when $T = \mathcal{O}(N^3)$. That is, their convergence rate can be $\mathcal{O}(1/\sqrt{NT})$ (linear speedup) only when $T = \mathcal{O}(N^3)$. In contrast, our analysis does not have such a requirement of the aggregation interval. Note that many other recent works have the same condition for linear speedup ($m^3 < K$) [2,3,4].
>
> Our analysis shows that FedLAMA guarantees the same complexity of convergence as FedAvg. In theory, the worst-case convergence speed of FedLAMA should be as fast as FedAvg with $\tau_{max}$. Therefore, we think our analysis provides a sufficiently solid performance guarantee.
>
> [1] Yu et al., On the Linear Speedup Analysis of Communication Efficient Momentum SGD for Distributed Non-Convex Optimization
>
> [2] Wang et al., Cooperative SGD: A Unified Framework for the Design and Analysis of Communication-Efficient SGD Algorithms
>
> [3] Wang et al., Tackling the Objective Inconsistency Problem in Heterogeneous Federated Optimization
>
> [4] Reddi et al., Adaptive federated optimization

---

> > ### Comment · Reviewer_7ks6 · 2021-11-15
> > **Lemma 5.2**
> >
> > For Lemma 5.2, $1-A$ should be greater than 0. There definitely should be a constraint for $\eta$.

---

> > > ### Author Response · Authors · 2021-11-15
> > > **Learning rate constraints.**
> > >
> > > Thank you for pointing out that! We acknowledge that Lemma 5.2 has the learning rate constraint as the reviewer pointed out ($A < 1$). We accordingly revised the paper and supplementary material. Consequently, Theorem 5.1. also has a slightly different learning rate constraint. We summarized this at the end of the proof of Theorem 5.1 in the supplementary material.

---

### Decision · Program_Chairs · 2022-01-20

**Decision:**

Reject

**Comment:**

This paper proposes a layer-wise adaptive aggregation method for federated learning that seeks to reduce the communication cost. The frequency of aggregation is adjusted separately for each layer of the model that is being trained. The number of iterations $\tau$ after which each layer's parameters are averaged across clients is multiplied by a factor $\phi$ depending on the magnitude of changes to the parameters at each layer. The paper gives a convergence analysis of the proposed method and provides experimental results to demonstrate its effectiveness in reducing communication without compromising accuracy.

Reviewer 7ks6 found some errors in the convergence analysis that were fixed by the authors in the discussion period. Reviewer YGZf increased their score to 5 after the discussion with the authors. The reviewers also gave the following suggestions to improve the paper:
1) Showing convergence curves to demonstrate that the communication reduction does not come at the cost of a slowdown in convergence.
2) The results presented in the paper shows only a small communication reduction for many of the layers. Perhaps the strategy can be improved in order to boost the communication reduction.
2) Use a more realistic model to calculate the communication cost so as to account for network delays and other costs rather than just considering the number of parameters communicated.

The scores are split on this paper. While one of the reviewers recommends acceptance, three others say that the paper is below the acceptance threshold. So I recommend a rejection while noting that the paper is close to the borderline.